# Evolutionary gain and loss of a plant pattern-recognition receptor for HAMP recognition

**Simon Snoeck[1], Bradley W Abramson[2], Anthony GK Garcia[1], Ashley N Egan[3], Todd P Michael[2], Adam D Steinbrenner[1]***

[1]Department of Biology, University of Washington, Seattle, United States; [2]The Plant Molecular and Cellular Biology Laboratory, Salk Institute for Biological Studies, La Jolla, United States; [3]Department of Biology, Utah Valley University, Orem, United States

**Abstract** As a first step in innate immunity, pattern recognition receptors (PRRs) recognize the distinct pathogen and herbivore-associated molecular patterns and mediate activation of immune responses, but specific steps in the evolution of new PRR sensing functions are not well understood. We employed comparative genomic and functional analyses to define evolutionary events leading to the sensing of the herbivore-associated peptide inceptin (In11) by the PRR inceptin receptor (INR) in legume plant species. Existing and de novo genome assemblies revealed that the presence of a functional *INR* gene corresponded with ability to respond to In11 across ~53 million years (my) of evolution. In11 recognition is unique to the clade of Phaseoloid legumes, and only a single clade of INR homologs from Phaseoloids was functional in a heterologous model. The syntenic loci of several non-Phaseoloid outgroup species nonetheless contain non-functional INR-like homologs, suggesting that an ancestral gene insertion event and diversification preceded the evolution of a specific INR receptor function ~28 my ago. Chimeric and ancestrally reconstructed receptors indicated that 16 amino acid differences in the C1 leucine-rich repeat domain and C2 intervening motif mediate gain of In11 recognition. Thus, high PRR diversity was likely followed by a small number of mutations to expand innate immune recognition to a novel peptide elicitor. Analysis of INR evolution provides a model for functional diversification of other germline-encoded PRRs.

**\*For correspondence:** astein10@uw.edu

**Competing interest:** The authors declare that no competing interests exist.

## Editor's evaluation

This manuscript, of interest to those studying the evolution of immunity, investigates the evolutionary history of a recently described herbivore-associated molecular pattern (HAMP) receptor, INR, which perceives the caterpillar-derived peptide HAMP, In11. The authors compare INR homologs to identify evolutionarily conserved residues and use chimeric fusion proteins to investigate specificity. The findings presented are valuable and supported by convincing experiments and analysis.

## Introduction

Plant and animal innate immunity is mediated by germline-encoded pattern recognition receptors (PRRs; *Ronald and Beutler, 2010*). PRRs serve as specific sensors of various pathogen-, herbivore-, and damage-associated molecular patterns (PAMPs, HAMPs, and DAMPs; *Snoeck et al., 2022*; *Albert et al., 2020*; *Gust et al., 2017*). In plants, PRRs in the large receptor kinase (RK) family comprise an extracellular domain, a single-pass transmembrane motif, and an intracellular kinase domain (*DeFalco and Zipfel, 2021*). Receptor-like proteins (RLPs) share structural similarities with RKs but lack a kinase

**eLife digest** The health status of a plant depends on the immune system it inherits from its parents. Plants have many receptor proteins that can recognize distinct molecules from insects and microbes, and trigger an immune response. Inheriting the right set of receptors allows plants to detect certain threats and to cope with diseases and pests.

Soybeans, chickpeas and other closely-related crop plants belong to a family of plants known as the legumes. Previous studies have found that, unlike other plants, some legumes are able to respond to oral secretions from caterpillars. These plants have a receptor known as INR that binds to a molecule called inceptin in the secretions. However, it remained unclear how or when INR evolved.

To address this gap, Snoeck et al. tested immune responses to inceptin in the leaves of 22 species of legume. The experiments revealed that only members of a subgroup of legumes called the Phaseoloids were able to recognize the molecule.

Analyzing the genomes of several legume species revealed that the gene encoding INR first emerged around 28 million years ago. Among the descendants of the legumes that first evolved this receptor, only the crop plant soybean and a few other species were unable to respond to inceptin. The genomic data indicated that these species had in fact lost the gene encoding INR over evolutionary time.

Snoeck et al. then combined data from genes encoding modern-day receptors to reconstruct the sequence of building blocks that make up the 28-million-year-old version of INR. This ancestral receptor was able to respond to inceptin in the caterpillar secretion, whereas an older version of the protein, which had a slightly different set of building blocks, could not. This suggests that INR evolved the ability to respond to inceptin as a result of small mutations in the gene encoding a more ancient receptor.

The work of Snoeck et al. reveals how the Phaseoloids evolved to respond to caterpillars, and how this ability has been lost in soybeans and other members of the subgroup. In the future, these findings may aid plant breeding or genetic engineering approaches for enhancing soybeans and other crops resistance to caterpillar pests.

domain (*Wang et al., 2008*; *Fritz-Laylin et al., 2005*). Extracellular domains of RK/RLPs include LysM, lectin, and malectin domains, but the most common ectodomain is a series of leucine-rich repeats (LRRs) which mediate PAMP binding and co-receptor association (*Albert et al., 2020*; *Shiu and Bleecker, 2003*; *Shiu and Bleecker, 2001*; *Fischer et al., 2016*; *Hohmann et al., 2017*; *Restrepo-Montoya et al., 2020*). In plants, LRR-RK/RLPs form a large gene family; for example, the *Arabidopsis thaliana* genome contains about 223 LRR-RKs and 57 LRR-RLPs (*Wang et al., 2008*; *Fritz-Laylin et al., 2005*; *Shiu and Bleecker, 2001*; *Lehti-Shiu et al., 2009*), and the number of annotated LRR-RK/RLPs per genome varies across plant species (*Restrepo-Montoya et al., 2020*; *Ngou et al., 2022*). Moreover, comparative genomic analyses of LRR-RK/RLPs involved in biotic interactions have revealed strong diversifying selection, lineage-specific expanded gene clusters, and immune receptor repertoire variation both within and between species (*Fritz-Laylin et al., 2005*; *Jamieson et al., 2018*; *Tang et al., 2010*; *Pruitt et al., 2021*; *Steinbrenner, 2020*), making the gene family a model for the evolution of innate immune systems.

Evidence is accumulating for highly specialized roles of plant LRR-RLPs as immune sensors (*Jamieson et al., 2018*; *Steinbrenner, 2020*). Various LRR-RLPs from *Arabidopsis*, tomato, wild tobacco, rapeseed, and cowpea have been shown to detect molecular patterns from fungi, bacteria, parasitic weeds, and herbivores (*Albert et al., 2020*; *Jamieson et al., 2018*). For example, Cf-9, Cf-4, and Cf-2 interact with respective molecular patterns of *Cladosporium fulvum*, Avr9, Avr4, and Avr2 (via host protein Rcr3), and are restricted to the *Solanum* genus (*Jones et al., 1994*; *Thomas et al., 1997*; *Kruijt et al., 2007*; *Dixon et al., 1996*; *Luderer et al., 2002*; *Krüger et al., 1979*; *Rooney et al., 2005*). *Arabidopsis* RLP42 recognizes fungal endopolygalacturonases (PGs) eptitope pg9(At) derived from *Botrytis cinerea* (*Zhang et al., 2014*; *Zhang et al., 2021*). Similarly, RLP23 is an *Arabidopsis*-specific LRR-RLP, which recognizes nlp20 peptide from the necrosis and ethylene-inducing peptide1 (NEP1)-like proteins (NLPs) found in bacterial/fungal/oomycete species (*Jamieson et al., 2018*; *Albert et al., 2015*). RXEG1 and RE02 were identified in wild tobacco and are respectively triggered by the fungal

elicitors XEG1 (*Phytophthora sojae*) and E02 (*Valsa mali*) (*Ma et al., 2015*; *Nie et al., 2021*). LepR3 (AvrLm1) and RLM2 (AvrLm2) convey both race specific resistance to the fungal pathogen *Leptosphaeria maculans* in rapeseed (*Larkan et al., 2015*; *Larkan et al., 2022*). ReMAX is restricted to the Brassicaceae and triggered by the PAMP eMax originating from *Xanthomonas* (*Jehle et al., 2013*). *Cuscuta* Receptor1 (*CuRe1*) is specific to *Solanum lycopersicum* and senses the peptide Crip21, which originates from parasitic plants of the genus *Cuscuta* (*Hegenauer et al., 2020*; *Hegenauer et al., 2016*). Finally, the inceptin receptor (INR) appears to be specific to the legume tribe Phaseoleae and recognizes inceptin (In11), a HAMP found in the oral secretions of multiple caterpillars (*Steinbrenner et al., 2020*; *Schmelz et al., 2006*; *Schmelz et al., 2012*; *Schmelz et al., 2007*). Notably, all the above examples of LRR-RLPs are family-specific, restricted to the Solanaceae (Cf-2, Cf-4, Cf-9, RXEG1, RE02, and CuRe1), Brassicaceae (RLP23, RLP42, LepR3, RLM2, and ReMAX), or Leguminosae (tribe Phaseoleae; INR; *Steinbrenner, 2020*; *Zhang et al., 2021*). However, despite clear signatures of lineage-specific functions, specific evolutionary steps leading to novel LRR-RLP functions across multiple species have not been described.

Mechanistic understanding of LRR-RLP sensing functions is also currently limited. Structural data exists for only one PAMP or HAMP sensing LRR-RLP, and genetic experiments have been limited to receptors from model species (*Zhang et al., 2021*; *Albert et al., 2019*; *Kourelis et al., 2020*; *Hohmann and Hothorn, 2019*; *Sun et al., 2022*). Ancestral sequence reconstruction (ASR) is a powerful approach for studying the evolution of protein function and specificity and has been applied to a limited number of receptor functions (*Delaux et al., 2019*; *Białas et al., 2021*; *Scossa and Fernie, 2021*; *Carroll et al., 2011*; *Chantreau et al., 2019*). However, ASR has not yet been applied to rapidly evolving LRR-RLPs functions, often encoded by complex multi-copy loci (*Steinbrenner, 2020*). Importantly, LRR-RLP recognition functions can be assessed through heterologous expression in a model plant, wild tobacco (*Nicotiana benthamiana*; *Zhang et al., 2021*; *Steinbrenner et al., 2020*; *Albert et al., 2019*; *Goodin et al., 2008*).

In this study, we use the legume-specific LRR-RLP INR as a model to perform dense species phenotyping, comparative genomics, and functional validation to associate gain and loss of HAMP response with evolution of the contiguous *INR* receptor locus. By leveraging both existing high-quality assemblies and long read, de novo assemblies of key legume species, we were able to analyze the ~53 million year (my) evolution of the *INR* locus. We show that In11 response is restricted to species in the ~28 my-old clade of the Phaseoloid legumes, which includes the agriculturally important subtribes Phaseolinae, Glycininae, and Cajaninae. Dynamic evolution of the syntenic *INR* receptor locus predates the divergence of Phaseoloids, but a single clade of *INR* homologs is active in recognition of In11. Finally, we used chimeric and ancestrally reconstructed LRR-RLPs to pinpoint key domains and amino acids (AA) involved in In11 recognition. Patterns of INR variation provide a framework for the evolution of novel functions within PRR gene families.

## Results

### In11 perception is restricted to certain legume species within the Phaseoloid clade

Response to the peptide HAMP In11 was previously thought to be highly restricted since only species in a subtribe of legumes, the Phaseolinae, which includes the species cowpea (*Vigna unguiculata*) and common bean (*Phaseolus vulgaris*), encode the corresponding INR receptor (*Steinbrenner et al., 2020*). To understand the emergence of INR function within legumes with a higher precision, we measured ethylene accumulation triggered by In11 as a defense marker across a set of 22 legume species of the NPAAA papilionoids (*Figure 1a*, *Supplementary file 5a, b*). Response to In11 was observed in plant species within the broader monophyletic clade of the Phaseoloids. Hence, phylogenetic evidence suggests a single origin of In11 response at the base of the Phaseoloid legumes ~28 my ago (mya; *Figure 1b*, ★). Several of the tested plant species/accessions within this clade were unable to respond to In11, namely *Hylodesmum podocarpum*, winged bean (*Psophocarpus tetragonolobus*), soybean (*Glycine max*), yam bean (*Pachyrhizus erosus*), and calopo (*Calopogonium mucunoides*), although these species were able to respond to the unrelated bacterial PAMP flg22 (*Figure 1—figure supplement 1*). These observations suggest the occurrence of multiple independent losses of In11 response throughout Phaseoloid evolution after the initial emergence of this function.

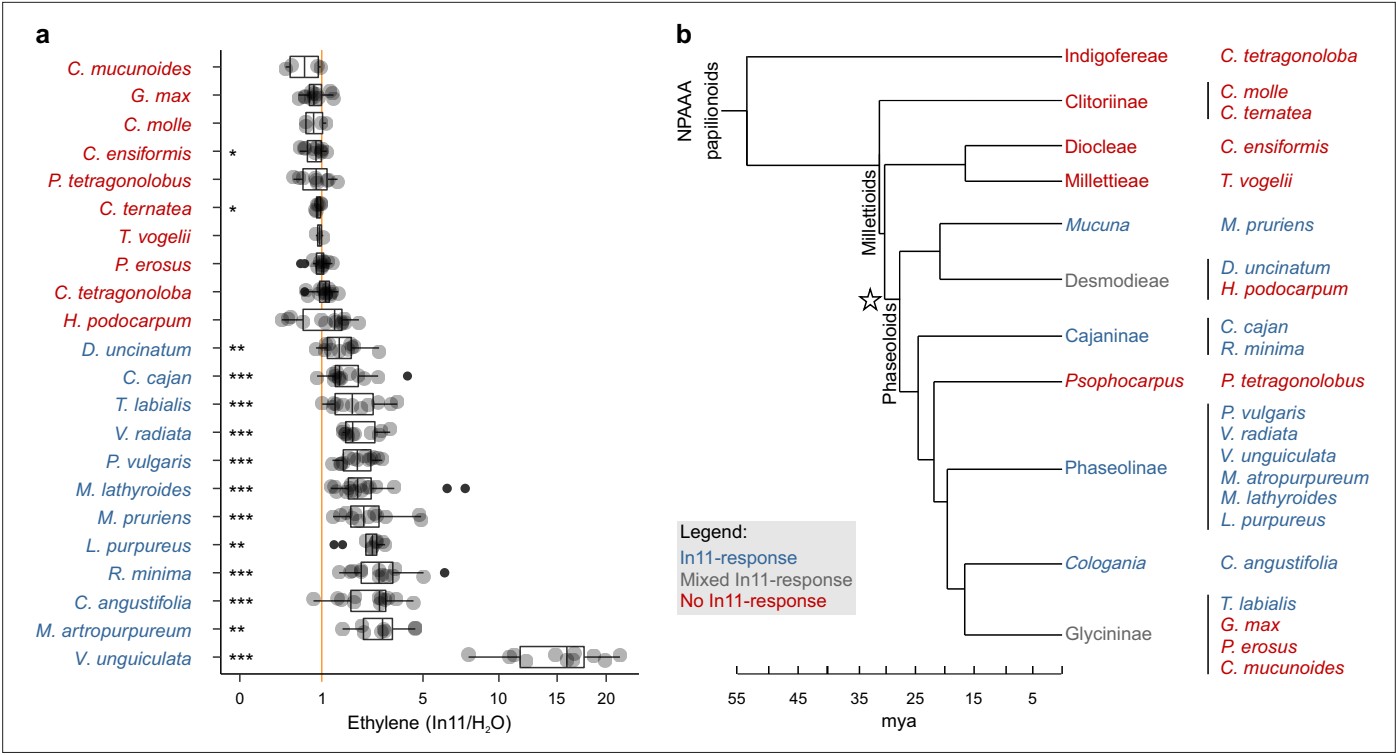

**Figure 1.** Induced ethylene response to In11 is limited to Phaseoloid legumes. (**a**) Individual trifoliate leaflets were scratch wounded and treated with 1 μM In11 or $H_2O$. The ratio of ethylene production for leaflets within the same leaf is shown (x-axis). The vertical orange line shows a ratio equal to one, i.e., no In11-induced ethylene burst. Biologically replicated plants are shown as separate dots. Significant differences between the control and the treatment of interest are indicated (paired Wilcoxon signed-rank test; ns non-significant, * $p \leq 0.05$, ** $p \leq 0.01$, and *** $p \leq 0.001$). Plant species names are colored blue (significant response to In11) or red (insignificant response). Species/accessions and resulting response data can respectively be found in **Supplementary file 5a** and **Supplementary file 5b**. (**b**) A summary chronogram representing time-based phylogenetic relationships within the tested lineages, with colors representing In11 response phenotypes as in (**a**). The star symbol indicates the node containing all In11-responsive species at the base of Phaseoloid legumes. Divergence time is shown in million years ago (mya) and represents a composite average (**Egan et al., 2016**; **Li et al., 2013**; **Lavin et al., 2005**; **Stefanović et al., 2009**).

The online version of this article includes the following figure supplement(s) for figure 1:

**Figure supplement 1.** Flg22-induced reactive oxygen species (ROS) and ethylene responses are idiosyncratic across Millettioid and non-Millettioid legume species.

## The contiguous *INR* locus shows LRR-RLP copy number variation in the Millettioids

To explore gain and loss of In11 response in relation to its defined PRR, we analyzed the evolution of its cognate receptor *INR* and its genomic locus in existing reference genomes and high quality de novo genome assemblies of Phaseoloid and non-Phaseoloid species. We focused de novo sequencing efforts on key nodes separating In11-responsive and unresponsive species in the phylogeny for comparative genomic analysis (**Figure 1b**). The contiguous *INR* locus, which is flanked by conserved anchor genes, was extracted from 16 existing legume genomes and four de novo assemblies: *P. erosus*, guar bean (*Cyamopsis tetragonoloba*), jack bean (*Canavalia ensiformis*), and *H. podocarpum* (**Figure 2**, **Supplementary file 6a**; **Steinbrenner et al., 2020**). For de novo assemblies obtained in this study, a combination of Oxford Nanopore and Illumina sequence data facilitated the assembly of a contiguous *INR* locus in all sequenced species (**Supplementary file 6b**).

The organization of the *INR* locus is highly diverse among legumes, with zero to seven LRR-RLP encoding-genes per species in addition to extensive repetitive sequence and transposable element content (**Figure 2a**, **Figure 2—figure supplement 1**, **Supplementary file 6c**). For all species, BLASTN search did not identify any regions with higher score than sequences at the contiguous *INR* locus of the respective species, strongly suggesting that all potential *INR* homologs are encoded at this syntenic locus. Legume species within the Hologalegina, namely *Lotus angustifolius*, chickpea (*Cicer*

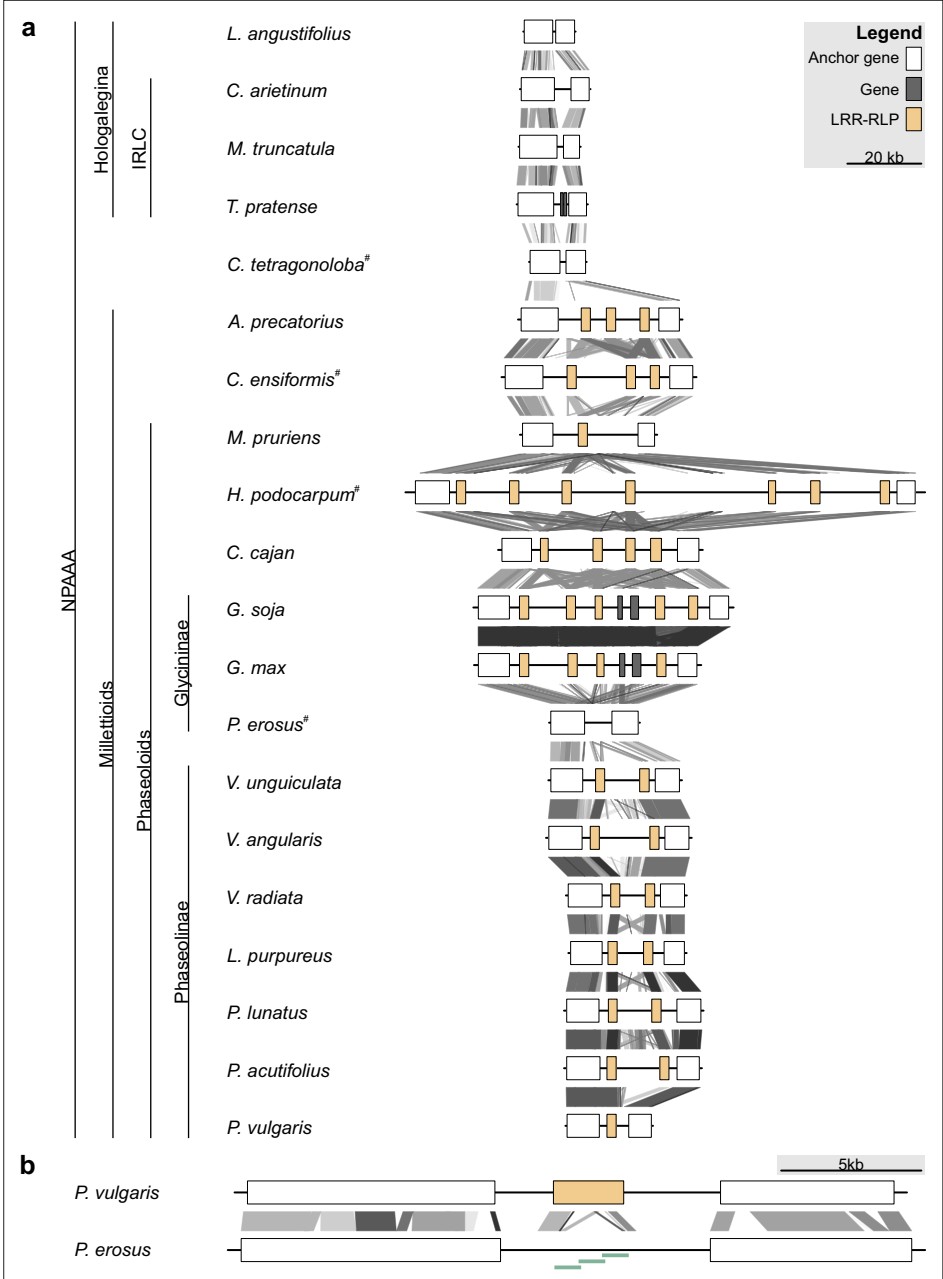

**Figure 2.** Leucine-rich repeat (LRR)-receptor-like protein (RLP) copy number variation at the *INR* locus in Millettioid and non-Millettioid legume genomes. Anchor, LRR-RLP, and other genes are colored as per legend. (**a**) Locus comparison of the contiguous *INR* locus of 20 NPAAA papilionoid species. Blast hits between loci are indicated with lines (e-value <1e-04) with score according to grayscale gradient and darker grays indicating higher similarity. Genes are labeled LRR-RLP if a complete coding sequence is present (≥875 AA). Species names followed by superscript (#) were newly sequenced and assembled for this project. (**b**) Locus comparison of the contiguous *INR* locus of *P. vulgaris* and *P. erosus*. *P. vulgaris* has a functionally validated inceptin receptor (INR) homolog (LRR-RLP) (*Steinbrenner et al., 2020*), while *P. erosus* lacks a full-length LRR-RLP. The disruption of the *P. erosus* LRR-RLP was validated by PCR followed by Sanger sequencing as indicated with light green bars.

The online version of this article includes the following figure supplement(s) for figure 2:

**Figure supplement 1.** Repetitive elements at the *INR* locus in Millettioid and non-Millettioid legume genomes.

**Figure supplement 2.** Prediction of gene duplication and gene loss events throughout the evolution of the contiguous *INR* locus in the Millettioids.

*arietinum*), *Medicago truncatula*, and clover (*Trifolium pratense*) do not contain an LRR-RLP encoding gene at the locus, whereas all species within the Millettioids except for *P. erosus* contained at least one LRR-RLP, consistent with a gene insertion event of an LRR-RLP at the *INR* locus in the ancestor of extant Millettioids (*Figure 1a*). To investigate this emergence, we analyzed the de novo assembly of *C. tetragonoloba*, a close outgroup of Millettioid legumes (*Figure 1b*; *Cardoso et al., 2015*; *Zhao et al., 2021*). As with all species in the outgroup Hologalegina, neither LRR-RLP was present between the anchor genes in *C. tetragonoloba* nor did we find an LRR-RLP with greater than 68% similarity in the whole genome, strengthening support for a single ancestral LRR-RLP gene insertion event ~32 mya (*Figure 2a*).

In contrast to other closely related legume species, BLASTN analysis of the contiguous *INR* locus of *P. erosus* revealed only partial coding sequence fragments with homology to LRR-RLPs, and consequently the absence of an *INR* homolog (*Figure 2b*). The de novo assembly of the locus was validated by performing PCR spanning the complete disruption and Sanger sequencing of the resulting amplicon. A Mariner-like transposase element exists between the LRR-RLP gene fragments (*Figure 2—figure supplement 1*, *Supplementary file 6c*). The absence of an *INR* homolog in *P. erosus* corresponds with species phenotype, namely the lack of induced ethylene response after In11 treatment (*Figure 1a*).

## In11-induced functions are conferred by a single clade of LRR-RLPs (*INR* clade)

To associate *INR* locus variation with the variable In11 responses (*Figure 1*), we next investigated the function and relationship of individual LRR-RLP homologs at the contiguous *INR* locus. We performed a maximum-likelihood phylogenetic analysis on the protein sequences of the LRR-RLPs within the locus across 16 Millettioid species (*Figure 2*). This analysis was supplemented with the closest related LRR-RLP genes outside the contiguous *INR* locus from *V. unguiculata* and *P. vulgaris*: *Phvul.007g246600* and *Vigun07g039700* (*Supplementary file 2*). A single well-supported clade which includes the previously characterized functional *INR* from *V. unguiculata* (*Vigun07g219600*) also contained orthologs from plant species able to respond to In11 (*Figure 3a*; *Steinbrenner et al., 2020*). Hence, we hypothesized that this clade contains functional *INR* homologs which can confer In11-induced functions.

To validate the putative *INR* clade, five genes (*Vigun07g219600*, *Phvul.007G077500*, *Mlathy INR*, *C.cajan_07316*, and *Mprur INR*) within this clade were cloned and transiently expressed in the non-legume model species *N. benthamiana*. Receptor function was measured using peptide-induced production of the plant defense markers ethylene and reactive oxygen species (ROS). In11-induced ethylene and ROS production were able to be conferred by each gene (*Figure 3b–c*), consistent with novel In11 recognition enabled by the *INR* clade (*Figure 3a*, blue labels). Intriguingly, cyclic ROS responses were observed for three out of five In11 responding receptors (Vigun07g219600, Phvul.007G077500, and Mlathy INR). Corresponding constructs without a C-terminal GFP tag reduced cyclic characteristics of the ROS response but resulted in equivalent ethylene bursts (*Figure 3—figure supplement 2*).

To assess if more distantly related, INR-like homologs could also confer In11 recognition, we measured ethylene and ROS production upon expression of LRR-RLPs outside the *INR* clade. Intriguingly, the sister clade to the *INR* clade contains LRR-RLPs from 9 out of 11 species which also have a predicted *INR* homolog in the *INR* clade itself (*Figure 3a*). Within this clade, we cloned and transiently expressed the LRR-RLP of *V. unguiculata* (*Vigun07g219700*). Except for *P. vulgaris* G19833, all studied Phaseolinae have an LRR-RLP in each of the two clades discussed above (*Figure 2*). The remaining species, *G. max*, *Glycine soja*, *H. podocarpum*, *C. ensiformis*, and *Abrus precatorius*, were In11-unresponsive (*Figure 1*) and solely encode LRR-RLP receptors which fall outside the *INR* clade and its sister clade. From this group, we cloned and transiently expressed the soybean LRR-RLP (*Glyma.10G228000*). Finally, we tested the most closely related LRR-RLP to *INR* outside the contiguous *INR* locus for both *V. unguiculata* and *P. vulgaris* (*Vigun07g039700* and *Phvul.007g246600*). No In11-induced responses could be observed upon heterologous expression of *Vigun07g219700*, *Glyma.10G228000*, *Vigun07g039700*, and *Phvul.007g246600* (*Figure 3b–c*). Proteins of these non-responsive LRR-RLPs were similarly or more strongly expressed in *N. benthamiana* relative to the lowest expressed responsive LRR-RLP (C.cajan_07316 INR; *Figure 3—figure supplement 1*), except for the marginally lower expressed Vigun07g039700, which contrasts with its closest related gene in

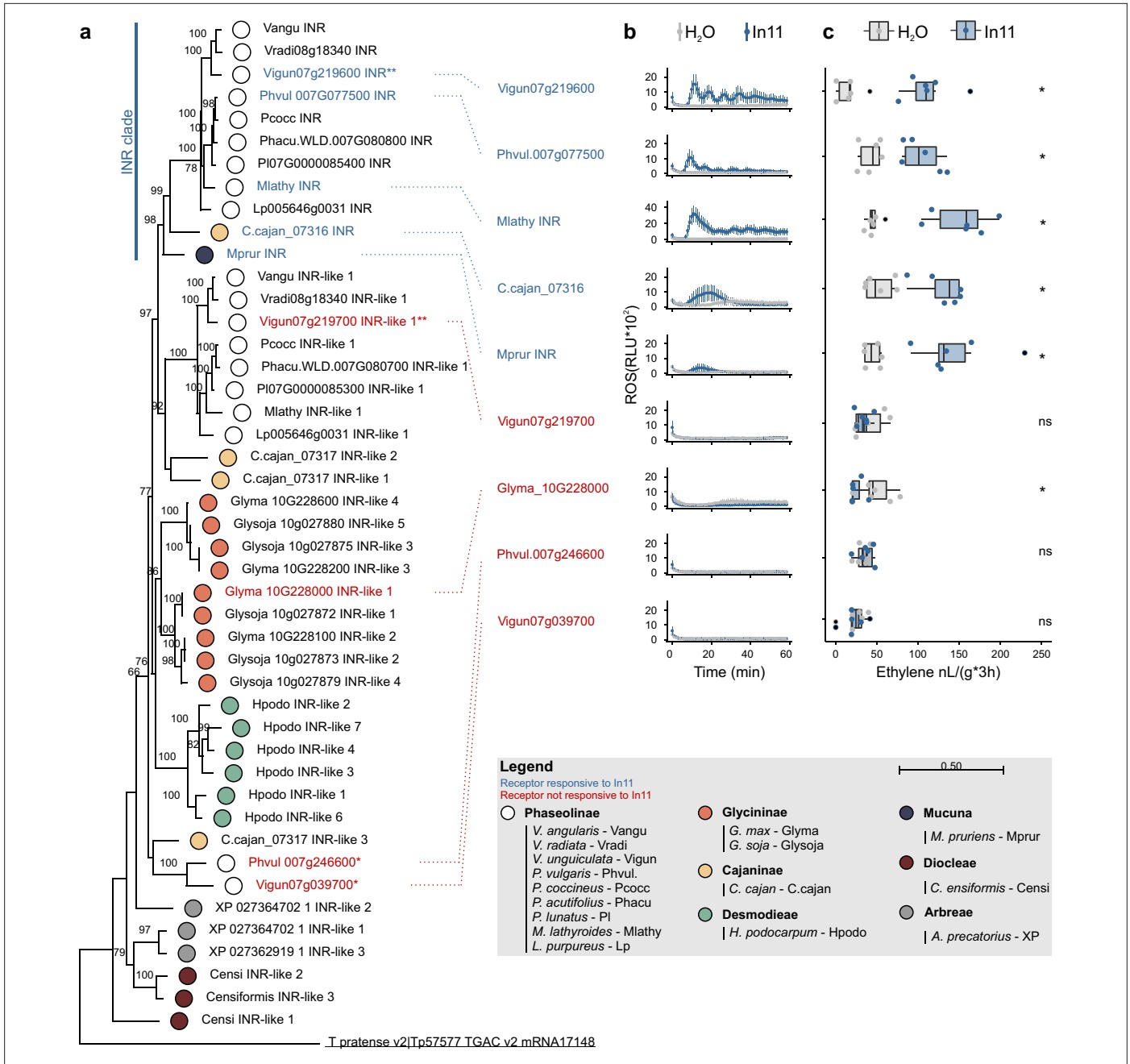

**Figure 3.** Phylogenetic analysis of leucine-rich repeat (LRR)-receptor-like proteins (RLPs) at the contiguous *INR* locus in 16 diverse species and subsequent heterologous expression reveals a clade of functional receptors. (**a**) A phylogenetic analysis of LRR-RLPs from 16 Millettioid species is shown. Maximum likelihood analysis bootstrap values are indicated, and only values higher than 65 are shown. The scale bar represents 0.5 AA substitutions per site. Filled dots indicate species of origin according to legend, where different colors indicate different subtribes. A *T. pratense* LRR-RLP was used as an outgroup to root the phylogenetic gene tree and is underlined. One asterisk (*) highlights the LRR-RLPs which are not part of a contiguous *INR* locus. Two asterisks (**) highlight the LRR-RLPs used to create the chimeric receptors. The functionally validated inceptin receptors (INRs) of *P. vulgaris*, *V. unguiculata*, *M. lathyroides*, *C. cajan*, and *M. pruriens* are highlighted in blue as they confer induced reactive oxygen species (ROS) and ethylene functions in response to In11 upon heterologous expression in *N. benthamiana*, as shown respectively in panel B and C. These five validated INRs fall within the labeled 'INR clade'. Heterologously expressed receptors which were not responsive to In11 are highlighted in red. (**b**) Shown are relative luminescence units (RLUs) after treatment with $H_2O$ (gray) or the peptide In11 (1 µM, blue). Curves indicate mean ± SD for four independent biological replicates (n=4 plants), with each biological replicate representing six technical replicates. (**c**) The x-axis shows the amount of ethylene released after infiltration with 1 µM In11 (blue) or water (gray). Dots represent independent biological replicates (n=6 plants). Significance was tested by performing a paired Wilcoxon signed-rank test (ns non-significant, * p≤0.05).

*Figure 3 continued on next page*

*Figure 3 continued*

The online version of this article includes the following source data and figure supplement(s) for figure 3:

**Figure supplement 1.** Western blot of the heterologously expressed constructs in *N. benthamiana*.

**Figure supplement 1—source data 1.** Original and labeled western blot of the heterologously expressed inceptin receptor (INR) and INR-like homolog constructs in *N. benthamiana*.

**Figure supplement 2.** The effect of a C-terminal GFP tag on In11-dependent reactive oxygen species (ROS) and ethylene burst of inceptin receptor (INR).

---

*P. vulgaris* (*Phvul.007g246600*, 87% AA similarity). Thus, the ability to confer In11-induced ROS and ethylene production is strictly limited to members of the *INR* clade.

We identified and dated potential gene duplications and losses at the contiguous *INR* locus throughout its evolution within the Millettioids using NOTUNG (*Figure 2—figure supplement 2*; *Chen et al., 2000*). Reconciliation analysis between gene and species trees revealed a complex evolutionary history comprising 19 gene duplications and 20 gene losses in total. Within the Millettioids, a single duplication event gave rise to the *INR* clade containing all *INR* homologs and its sister clade which contains the closest related *INR*-like homologs. *P. vulgaris* and *Mucuna pruriens* contain an *INR* clade homolog but not members of the sister *INR*-like 1 clade, consistent with two independent gene losses. In contrast, the analysis predicts the ancestral loss of an *INR* homolog within the *INR* clade for *G. soja*, *G. max*, and *H. podocarpum* and clade specific duplication events resulting in LRR-RLP expansions of 4–7 tandem duplicates. To confirm that *INR* was lost in *Glycine* and not just in reference assemblies, we performed BLASTP searches using Vigun07g219700 AA sequence against 26 *G. soja* and *G. max* de novo assemblies from a recent pangenome analysis (*Liu et al., 2020*). Like the *Glycine* reference genomes, no BLASTP hits to Vu07g219600 for any of the 26 de novo assemblies were identified that had an AA similarity higher than 76%, suggesting that INR was lost before the speciation of *G. max* and *G. soja*, i.e., prior to soybean domestication.

## The C1 and C2 subdomain of the LRR ectodomain mediate In11-induced functions

To understand receptor subdomains contributing to the functional *INR* clade, we assembled nine chimeric receptors combining *Vigun07g219600* (*Vu*INR hereafter) and paralogous *Vigun07g219700* (*Vu*INR-like, 72% AA similarity; *Supplementary file 3*). Both genes contain a typical LRR-RLP extracellular domain with 29 LRRs interrupted by a 16-AA intervening motif (C2 domain) (*Figure 4*; *Steinbrenner et al., 2020*). Chimeric receptors were expressed at a similar level in *N. benthamiana*, and their response to In11 was assessed by quantifying peptide-induced ethylene and ROS (*Figure 4*, *Figure 4—figure supplement 1*). Chimeric receptors with the *Vu*INR-like C1 domain were not responsive to In11. In contrast, all chimeric receptors containing the C1-C2 of *Vu*INR responded to In11 treatment with both an ethylene and ROS burst (219600 F, 219600-C3, 219600-C2, and C1-219600; *Figure 4a*). Intriguingly, the chimeric receptor 219600−C1, containing C1 of *Vu*INR and C2 of *Vu*INR-like, responded to the control treatment. Nonetheless, no visible phenotype was observed 7 days post-infiltration of 219600-C1 in *N. benthamiana* with or without subsequent infiltration of flg22, In11, or H$_2$O 1 day post the construct infiltration (*Figure 4—figure supplement 2a–2b*). Additionally, the chimeric receptor, 219600−C2, responds to In11 but has a delayed ROS burst relative to all other In11-responsive constructs tested. In summary, the *Vu*INR LRR subdomains C1 and C2 are both required for In11 response in chimeric receptors.

## ASR of the *INR* LRR domain

To further understand the molecular basis for functional divergence between INR and unresponsive INR-like homologs, we predicted multiple ancestral sequences of the monophyletic In11-responsive *INR* clade and its non-responsive *INR*-like 1 sister clade (*Figure 5*, *Figure 5—figure supplement 1*, *Supplementary file 4*). To perform ASR, we first confirmed that nodes of interest for reconstruction were well supported by both neighbor-joining (NJ) and maximum likelihood (ML) gene phylogenies (*Figure 4—figure supplement 1*). Subsequently, we reconstructed the ancestral sequences for the LRR domain using FastML (*Supplementary file 4*; *Randall et al., 2016*), synthesized their predicted sequences, and ligated the resulting LRR domains with flanking domains (A–B and D–G) from *Vu*INR

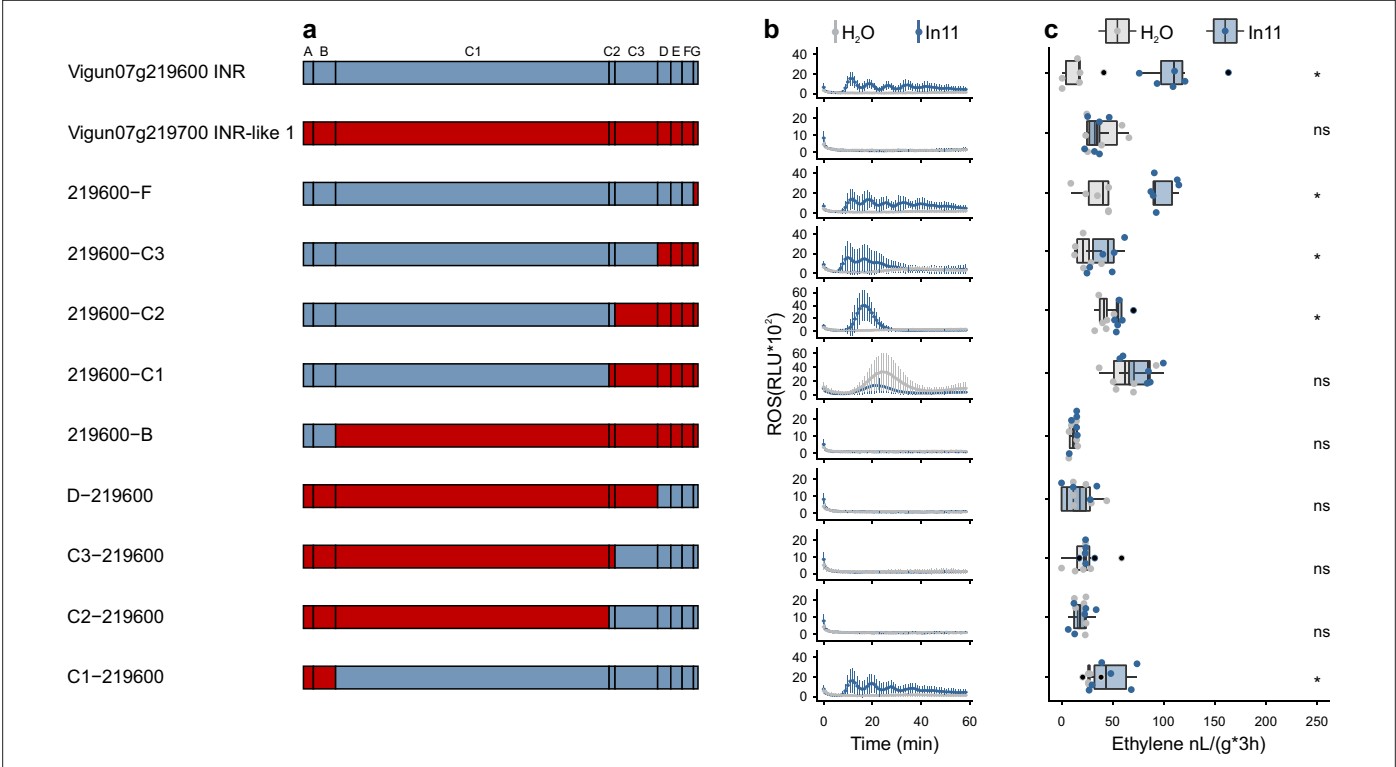

**Figure 4.** Chimeric receptors indicate that the C1 and C2 subdomains mediate inceptin receptor (INR) recognition function. (**a**) Schematic representation of *Vigun07g219600* (blue, *Vu*INR), *Vigun07g219700* (red, *Vu*INR-like), and the nine created chimeric receptors used for structure-function analysis. Leucine-rich repeat (LRR)-receptor-like protein (RLP) subdomains as per ***Fritz-Laylin et al., 2005***; A: putative signal peptide, B: one or two pairs of Cys that may play structural roles, C: multiple LRRs with an intervening motif (**C2**) inserted, D: linker domain, E: acidic domain, F: transmembrane helix, and G: cytoplasmic tail. Nucleotide sequences of the chimeric receptors can be found in ***Supplementary file 3***. (**b**) In11-dependent reactive oxygen species (ROS) production following the heterologous expression of receptors in *N. benthamiana*. Shown are relative luminescence units (RLUs) after treatment with $H_2O$ (gray) or the peptide In11 (1 µM, blue). Curves indicate mean ± SD for four independent biological replicates (n=4 plants), with each biological replicate representing six technical replicates. (**c**) Ethylene production following the heterologous expression of receptors in *N. benthamiana*. Ethylene production was quantified after infiltration with $H_2O$ (gray) or the peptide In11 (1 µM, blue). Dots represent independent biological replicates (n=6 plants). Significance was tested by performing a paired Wilcoxon signed-rank test (ns non-significant, * p≤0.05).

The online version of this article includes the following source data and figure supplement(s) for figure 4:

**Figure supplement 1.** Western blot of the heterologously expressed chimeric receptor constructs in *N. benthamiana*.

**Figure supplement 1—source data 1.** Original and labeled western blot of the heterologously expressed chimeric receptor constructs in *N. benthamiana*.

**Figure supplement 2.** Heterologous expression of chimeric receptor 219600-C1 does not result in a visible phenotype in *N. benthamiana*.

to complete the ancestral receptor constructs (labeled in ***Figure 5a***). Protein expression in *N. benthamiana* was similar across all ASR variants (***Figure 5—figure supplement 2***).

Ancestral receptors to the *INR* clade of the *Vigna*, *Phaseolus*, and *Macroptylium* (N7), the Phaseolinae (N6) and the Phaseoloids (N4) conferred In11-induced ethylene and ROS response. In contrast, ancestral receptors of the *INR*-like 1 clade of the Phaseolinae (N16) and the Phaseoloids (N14) were not responsive to In11 (***Figure 5d***, ***Figure 5c***). Moreover, the common ancestral receptor of the *INR* and its *INR*-like 1 sister clade (N3) is not responsive to In11 (***Figure 5***). Hence, a small number of differences, 16 of 720 AA, between the LRR domain of N3 and N4, mediate differential In11 response. Moreover, only a subset of those AAs is similar within all LRR-RLPs of the *INR* clade and different in LRR-RLPs of the *INR*-like 1 sister clade: Y91, H239, A359, H404, and R406 (***Supplementary file 8***). Thus, reconstruction of ancestral *INR* homologs pinpoints a limited number of SNPs which mediate In11 recognition.

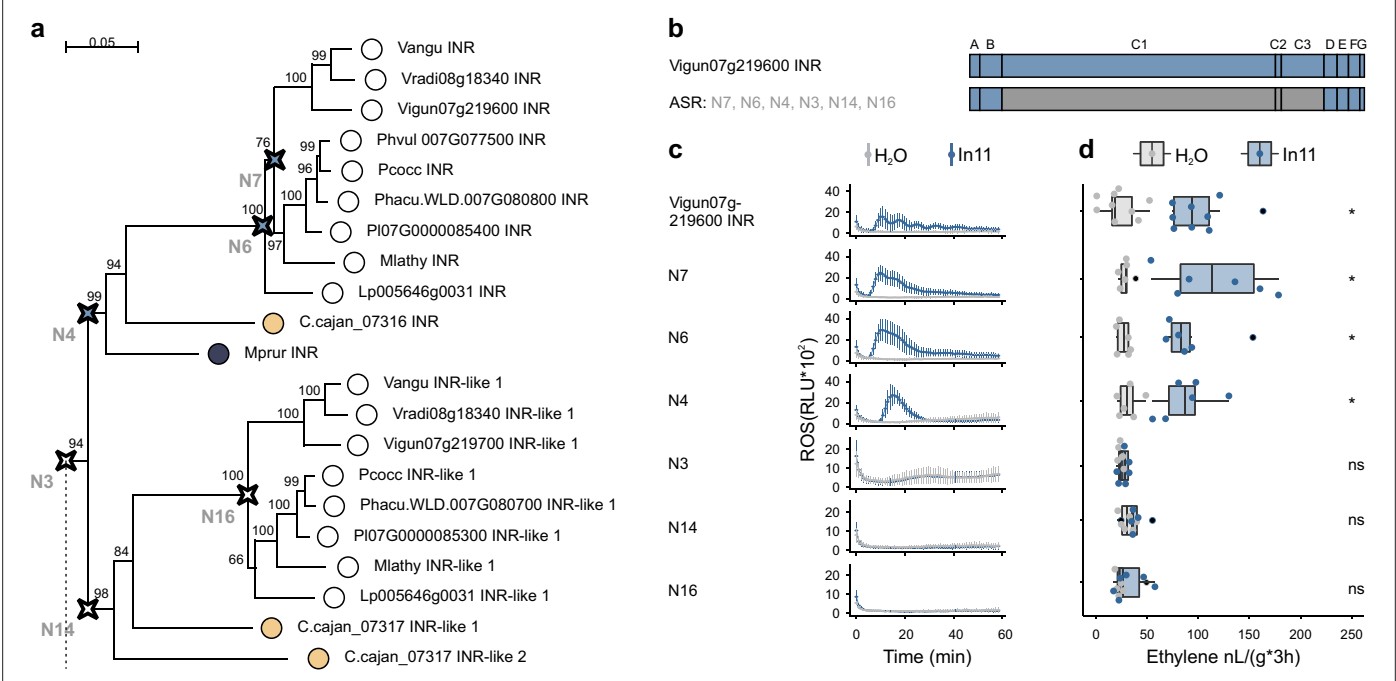

**Figure 5.** Functional analysis of ancestrally reconstructed leucine-rich repeat (LRR) domains of inceptin receptor (INR) and INR-like receptors. (**a**) Part of the phylogenetic analyses of the LRR (C1-3 domain) of LRR-receptor-like proteins (RLPs) from the contiguous *INR* locus used for the ancestral sequence reconstruction (ASR) full phylogeny can be found in *Figure 5—figure supplement 1*. Ancestrally reconstructed nodes are marked with a compass star (**N7, N6, N4, N3, N14, and N16**). The scale bar represents 0.05 AA substitutions per site. Nucleotide sequences of the LRR domain can be found in *Supplementary file 4*. (**b**) Schematic representation of *Vigun07g219600* (blue, *Vu*INR) and the six created ASR receptors used for structure-function analysis. LRR-RLP subdomains as per *Fritz-Laylin et al., 2005*; A: putative signal peptide, B: one or two pairs of Cys that may play structural roles, C: multiple LRRs with an intervening motif (**C2**) inserted, D: linker domain, E: acidic domain, F: transmembrane helix, and G: cytoplasmic tail. Nucleotide sequences of the chimeric receptors can be found in *Supplementary file 3*. (**c**) In11-dependent reactive oxygen species (ROS) production following the heterologous expression of the ASR receptors in *N. benthamiana*. Relative luminescence units (RLUs) are shown after treatment with $H_2O$ (gray), or the peptide In11 (1 µM, blue). Curves indicate mean ± SD for four independent biological replicates (n=4 plants), with each biological replicate representing six technical replicates. (**d**) Ethylene production following the heterologous expression of receptors in *N. benthamiana*. Ethylene production was quantified after infiltration with $H_2O$ (gray) or the peptide In11 (1 µM, blue). Dots represent independent biological replicates (n≥6 plants). Significance was tested by performing a paired Wilcoxon signed-rank test (ns non-significant, * p≤0.05).

The online version of this article includes the following source data and figure supplement(s) for figure 5:

**Figure supplement 1.** Phylogenetic analyses of the leucine-rich repeat (LRR) (C1-3 domain) of LRR-receptor-like proteins (RLPs) from the contiguous *INR* locus.

**Figure supplement 2.** Western blot of the heterologously expressed ancestral sequence reconstruction (ASR) receptor constructs in *N. benthamiana*.

**Figure supplement 2—source data 1.** Original and labeled western blot of the heterologously expressed ancestral sequence reconstruction (ASR) receptor constructs in *N. benthamiana*.

## N404H and K406R AA substitutions abolish the function of N4, the In11-responding ancestor of all INR-homologs

To pinpoint specific AA mediating In11 recognition, we created N4 constructs with single AA substitutions to the corresponding AA of N3: Y91F, H239D, A359S, H404N, and R406K (*Supplementary file 8*, *Figure 6*). These AA are the only N3 vs N4 polymorphisms with 100% similarity within the INR-clade, and we reasoned that these might be important for INR and N4 function. N4 and mutated N4 receptors were expressed at a similar level in *N. benthamiana*, and their response to In11 was assessed by quantifying peptide-induced ethylene and ROS (*Figure 6—figure supplement 1*). N4 H239D and N4 A359S responded to In11 treatment with both an ethylene and ROS burst (*Figure 6c–d*). Similarly, N4 Y91F responded to In11 treatment although with a relatively lower ROS burst. The position of Y91F AA substitution is intriguing as it might be involved in the potential interaction with N-terminal C1-domain and the C2-domain as suggested by an AlphaFold prediction (*Figure 6b*). Finally, N4 H404N and

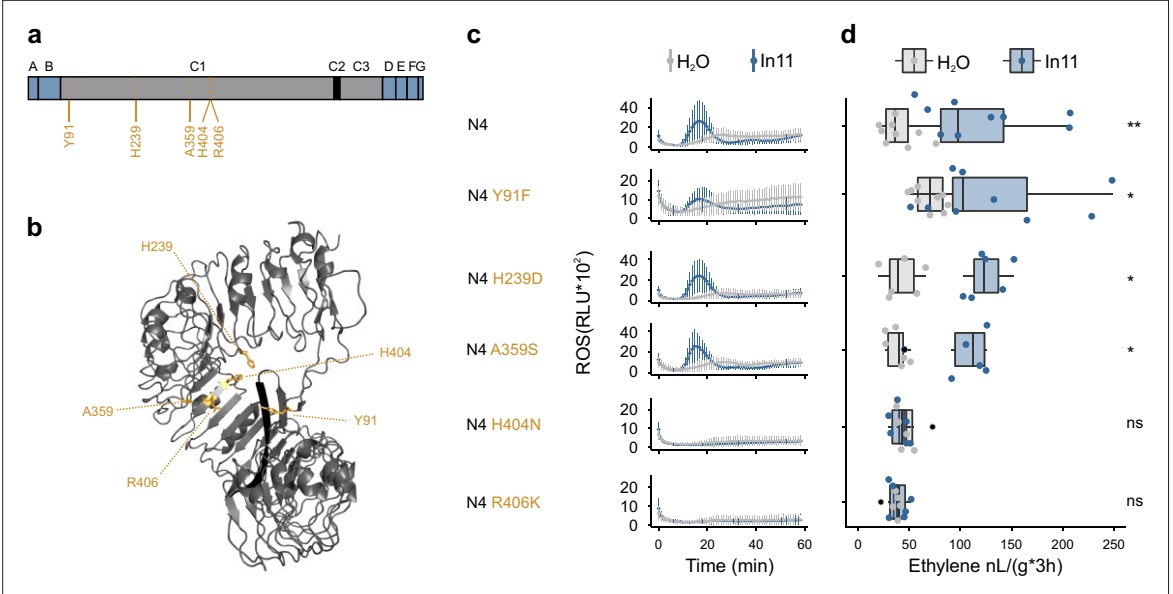

**Figure 6.** The H404N and R406K AA substitutions abolish the function of N4, the ancestor of all inceptin receptor (INR)-homologs. (**a**) Schematic representation of the N4 ancestral sequence reconstruction (ASR) receptor used for structure-function analysis. Five strongly conserved amino acid (AA) in all INR-homologs and N4 which differ from the corresponding N3 AA are highlighted in yellow. Leucine-rich repeat (LRR)-receptor-like protein (RLP) subdomains as per *Fritz-Laylin et al., 2005*; A: putative signal peptide, B: one or two pairs of Cys that may play structural roles, C1: multiple LRRs (colored in gray) with an intervening motif (C2, colored in black) inserted, D: linker domain, E: acidic domain, F: transmembrane helix, and G: cytoplasmic tail. (**b**) AlphaFold prediction of *Vu*INR (Vigun07g219600), only the C-domain is depicted with the C2-domain colored in black. Similar as in (**a**), the five strongly conserved AAs in all INR-homologs and N4 which differ from the corresponding N3 AAs are highlighted in yellow. (**c**) In11-dependent reactive oxygen species (ROS) production following the heterologous expression of the N4 and N4 variant receptors in *N. benthamiana*. Relative luminescence units (RLUs) are shown after treatment with $H_2O$ (gray) or the peptide In11 (1 μM, blue). Curves indicate mean ± SD for four independent biological replicates (n=4 plants), with each biological replicate representing six technical replicates. (**d**) Ethylene production following the heterologous expression of receptors in *N. benthamiana*. Ethylene production was quantified after infiltration with $H_2O$ (gray) or the peptide In11 (1 μM, blue). Dots represent independent biological replicates (n≥6 plants). Significance was tested by performing a paired Wilcoxon signed-rank test (ns non-significant, * p≤0.05).

The online version of this article includes the following source data and figure supplement(s) for figure 6:

**Figure supplement 1.** Western blot of the heterologously expressed N4 and N4 variant constructs in *N. benthamiana*.

**Figure supplement 1—source data 1.** Original and labeled western blot of the heterologously expressed N4 and N4 variant constructs in *N. benthamiana*.

**Figure supplement 2.** Predicted LDDT(Local distance difference test) score per position for all five created AlphaFold models of Vigun07g219600.

N4 R406K did not respond to In11, allowing us to pinpoint to two AA substitutions which appeared to be crucial in the evolution of INR function.

## Discussion

Specificity of PRR recognition functions underlies innate immunity in plants and animals (*Ronald and Beutler, 2010*). Here, we described the evolution of a lineage-specific recognition function of the legume-specific LRR-RLP INR. We identified specific steps in INR evolution: an ancestral gene insertion event, receptor gene diversification, and the evolution of specific INR recognition capability. Finally, comparisons between In11-responsive and unresponsive homologs, especially through chimeric receptors, ASR, and single AA substitutions, pinpoint key AA residues which mediate peptide recognition and response. Our work illuminates themes in the evolution of lineage-specific immune sensing functions, which will inform the broad use of PRRs as resistance traits in agriculture (*Kourelis et al., 2020*; *Schultink and Steinbrenner, 2021*; *Lacombe et al., 2010*; *Pfeilmeier et al., 2019*).

Evolutionary analysis of diverse plant immune responses to a single PAMP or HAMP is rarely performed beyond a single model species, although this can be a powerful approach to understand the emergence of specific immune receptor functions (*Snoeck et al., 2022*). We measured In11 responses

across 22 legume species, indicating that In11 response is restricted to the Phaseoloids (*Figure 1*). However, within the Phaseoloids, several tested species/accessions were not able to respond to In11, suggesting multiple independent losses of INR. These phenotypic observations allowed the investigation of the evolution of a specific LRR-RLP involved in plant immunity over a long (~32 my), detailed timescale.

We complemented broad analysis of In11 response with comparative genomics of key legume species within the NPAAA papilionoid clade, which diverged ~53 mya (*Egan et al., 2016*; *Li et al., 2013*; *Lavin et al., 2005*; *Stefanović et al., 2009*). Analysis of the *INR* receptor locus across 20 existing and newly sequenced legume genomes revealed gene insertion and diversification at the *INR* locus in Millettioid species, which diverged prior to the Phaseoloids and consequently the emergence of In11 recognition. The *INR* locus shows high variability and diversification among the larger group of Phaseoloid and non-Phaseoloid legumes of the NPAAA papilionoids, with zero to seven LRR-RLP encoding-genes per species. Legume species within the Hologalegina (*L. angustifolius*, *C. arietinum*, *M. truncatula*, and *T. pratense*) do not contain an LRR-RLP encoding gene at the locus, whereas all species investigated herein the Millettioids except for *P. erosus* contain at least one LRR-RLP (*Figure 2*). This observation is consistent with a gene insertion event at the *INR* locus in the ancestor of extant Millettioids ~32 mya, which likely led to extant copy number variation in early-branching Millettioid species. Processes generating receptor diversity in high copy number variation loci will be interesting to explore for model PRRs including INR (*Krasileva, 2019*).

Analysis of INR provides an example of a PRR with conserved recognition function across a relatively long evolutionary history (since the emergence of the Phaseoloids ~ 28 mya), compared to other extensive studied LRR-RLPs which are generally studied within a single genus or species (*Albert et al., 2019*; *Kourelis et al., 2020*). Each of 364 tested varieties of *V. unguiculata* are able to respond to In11 and our analysis now extends this conservation across legume species (*Steinbrenner et al., 2020*). A phylogenetic analysis revealed potential INR-homologs in 10 additional species as they clustered together with *V. unguiculata* INR. Orthologous LRR-RLPs of the *INR* clade from five legume genera were tested and able to induce an In11-induced ethylene and ROS burst upon heterologous expression in *N. benthamiana* (*Figure 3*). This includes INR homologs from *Cajanus cajan* and *M. pruriens* of the Phaseoloid clade. Certain Phaseoloids and earlier diverged non-Phaseoloids also contain INR-like homologs at the contiguous *INR* locus; thus, INR most likely arose from an existing LRR-RLP, ~28 mya. The role of INR-like homologs is not clear, but they may detect a related, In11-like ligand. This evolutionary pattern is reminiscent of the evolution of LRR-RLP Cf-2 which evolved <6 mya within the genus *Solanum*, potentially by intergenic recombination (*Kourelis et al., 2020*). In summary, an ancestral gene insertion event is the likely source of the extant gene and copy number variation at the INR locus, which preceded the evolution of a specific peptide recognition function. The high level of AA conservation of the recognized ligand In11 (*Schmelz et al., 2006*) may drive overall stability of the INR locus since the divergence of the Phaseoloids ~28 mya.

The abundance of closely related legume genomes provided by this study and others reveals the dynamic nature of *INR* receptor loci (*Chen et al., 2000*; *Shi et al., 2014*). Interestingly, we observed at least two independent cases of *INR* loss within the Phaseoloids. Two reference genomes as well as 26 de novo assemblies of the closely related Glycininae, *G. soja* and *G. max*, do not encode *INR* (*Liu et al., 2020*). Nevertheless, *INR* loss does not seem to predate the divergence of the Glycininae as *Teramnus labialis* is able to respond to In11 (*Figure 1*), although genome data are lacking for this species. Besides reciprocal gene loss of *INR*, gene tree-species tree reconciliation also suggests the involvement of five tandem duplications of an *INR*-like gene predating radiation of the genus *Glycine* (*Figure 2—figure supplement 2*). This is consistent with whole-genome observations of preferential gene loss in tandem clusters (*Shi et al., 2014*). Similarly, within the separate Desmodieae lineage, our analysis also predicts that a *H. podocarpum* INR-like gene underwent specific tandem duplication events after the loss of *INR*, as its genome contains seven *INR*-like homologs which cluster together in a phylogenetic analysis. In contrast, *P. vulgaris* and *M. pruriens* lost *INR*-like, and *P. erosus* seems to have lost both *INR* and *INR*-like as it does not contain any complete coding sequence of an LRR-RLP at the *INR* locus. Repetitive elements and transposable elements within the INR locus may have disrupted ancestral INR and INR-like genes in these lineages and can be found between LRR-RLP genes and gene fragments (*Figure 2—figure supplement 2*). Loss of *INR* (and *INR*-like) may reflect

the propensity of tandemly duplicated loci to lose functions through gene conversion and/or reciprocal gene loss (*Shi et al., 2014*).

Our detailed analysis of the emergence of INR provides a roadmap for understanding evolution of recognition specificity for other lineage-specific PRRs (*Steinbrenner, 2020*; *Zhang et al., 2021*). For analogous RLP-mediated responses within the Brassicaceae, extensive phenotyping was earlier performed for several PAMPs – eMax, nlp20, SCFE1, IF1, and pg13 – across *Arabidopsis* accessions and related species (*Zhang et al., 2021*; *Albert et al., 2015*; *Jehle et al., 2013*; *Shi et al., 2014*; *Wei et al., 2020*). However, analysis of the respective PRR loci across Brassicaceae has not yet been conducted and may reveal processes leading to their dynamic evolution. A recent pangenomic analysis of *Arabidopsis* indicates that while many RLPs occur in complex or copy-number variable loci (*Steinbrenner, 2020*; *Zhang et al., 2021*), three PAMP sensing LRR-RLPs (RLP 23, RLP30 and RLP32) are conserved across *A. thaliana* varieties (*Pruitt et al., 2021*). Further cross-species analysis may reveal a similar trajectory to INR via ancestral duplications preceding fixation in the *Arabidopsis* lineage. In summary, additional case studies for specific receptors are needed to reveal broader patterns in PRR evolution.

Our functional analysis of INR and INR-like genes in a heterologous model (*N. benthamiana*) also provides insight into LRR-RLP function. Chimeric receptors formed by combining *INR* and an *INR*-like paralog revealed LRR-RLP subdomains required for In11 response. Across all chimeric receptors, those encoding the LRR (C1) and intervening motif (C2) domains of *Vu*INR responded to In11 treatment with both an ethylene and ROS burst, suggestive of crucial elements for elicitor interaction in the C1-C2 subdomain. Notably, a chimeric receptor with mixed C1 and C2, 219600-C1, is the sole construct that responded to the control treatment, suggestive of a critical role for C1-C2 compatibility. In addition, a chimeric receptor with mixed C2 and C3 (219600-C2) had delayed In11-induced ROS burst relative to all other In11-responsive constructs tested here (*Figure 4*). These findings are consistent with previous truncation and chimeric protein analyses for *Arabidopsis* RLP23 and RLP42. Truncations of the RLP23 ectodomain abolished function, suggesting the necessity of the entire ectodomain for elicitor binding or proper assembly of the receptor (*Albert et al., 2019*). Additionally, chimeric receptors implicated the importance of RLP42 its 12 N-terminal LRRs (C1) and LRR21-LRR24 which includes the intervening motif (C2) for recognition of fungal PG (*Zhang et al., 2021*). Furthermore, RXEG1, the only PAMP or HAMP sensing LRR-RLP with structural information interacts primarily with its ligand XEG1 through two loopout regions, one in the N-terminal C1 domain and another comprising the C2 domain (*Sun et al., 2022*). Both the C1 domain and the C2 intervening motif are therefore critical for LRR-RLPs in multiple plant families, but their specific role in PAMP/HAMP recognition needs further investigation.

Additionally, INR functional analyses are consistent with a vestigial role for the cytoplasmic tail of PRRs in the LRR-RLP family. Strikingly, all identified INR homologs encode a cytoplasmic tail of only 10 AA, shorter relative to all identified INR-like homologs here. Nevertheless, swapping the *Vu*INR (Vigun07g219600) cytoplasmic tail to the extended version of *Vu*INR-like 1 (Vigun07g219700), 219600 F, did not affect In11-response (*Figure 4*). Previously, the complete deletion of the intracellular 17-amino-acid tail of RLP23 reduced but did not abolish receptor function (*Albert et al., 2019*). Additionally, a previously described chimeric swap replacing the RLP42 terminal LRR, transmembrane helix, and cytoplasmic tail with the respective subdomains of a non-responsive paralog was still responsive to the pg9(At) elicitor (*Zhang et al., 2021*). Intriguingly, quantitative differences were earlier reported as the C-terminally truncated RLP23 had a reduced nlp20 response, consistent with auxiliary rather than essential function for the RLP23 cytoplasmic tail (*Albert et al., 2019*).

In addition to the use of chimeric receptors, reconstruction of the ancestral *INR* LRR domain revealed further functional insights contributing to INR function. Our analysis suggests that the common ancestral LRR ectodomain of extant INRs conferred In11 response (N4), whereas the common ancestor of both INR and INR-like (N3) did not confer In11 response (*Figure 5*). N3 and N4 differ in In11 response although they only vary by 16 AA, and 5 of the 16 show strongly conserved (100% AA similarity) across all members of the extant INR clade (*Supplementary file 8*). Two of five single AA substitutions in N4 to the corresponding N3 AA, H404N and R406K, abolished In11-induced responses (*Figure 6*). Thus, these AA substitutions and change in sidechain charge (N to H) were critical for the evolution of the novel recognition functions of INR. The role of key LRR-RLP residues in ligand-specific responses is consistent with previous single AA substitutions to the functional RLP42 receptor, where several were sufficient to abolish pg9(At)-induced co-receptor association and defense responses (*Zhang et al.,*

*2021*). Hence, based on studies of LRR-RLP function, evidence is accumulating for the critical roles of specific residues for several LRR-RLPs in both C1 and C2 domains.

A similar ASR approach was previously employed to understand an effector recognition domain, the 98-AA heavy metal-associated (HMA) domain of the plant intracellular NOD-like receptor Pik-1 (*Białas et al., 2021*). Specific AA changes introduced into an ancestrally reconstructed backbone were sufficient to confer effector binding and immune functions. For Pik-1 HMA, structural data provided additional insight into the mechanism of effector binding in ancestral and extant proteins. Our use of an ASR approach for a relatively long LRR domain (720 AA) now demonstrates the power of dense comparative genomic analyses to also identify key residues in extant PRRs without defined binding sites. In the absence of structural data for ligand-binding LRR-RLPs, an ASR approach may be useful to identify sets of co-varying residues critical for binding and signaling functions of PRRs.

## Materials and methods
### Plant materials
Plant species and accessions used in this study are listed in *Supplementary file 5*a, as well as their respective providers: Phil Miklas, (US Department of Agriculture, Prosser, WA, USA), Creighton Miller (Texas A&M University, College Station, TX, USA), Phil Roberts (UC Riverside, CA, USA), Timothy Close (UC Riverside, CA, USA), and the USDA Germplasm resources Information network (GRIN). Plants were grown in the greenhouse (25/21°C day/night, 60% RH(relative humidity) and 12:12 light:dark cycles) or in growth chambers (26/26°C day/night, 70% RH and 12:12 light:dark cycles).

### Peptide-induced ethylene production in legume species
The In11 peptide (sequence ICDINGVCVDA) is a host derived proteolytic fragment of the ATP synthase γ-subunit (cATPC), based on the *V. unguiculata* cATPC sequence (*Schmelz et al., 2006*). The flg22 peptide (sequence QRLSTGSRINSAKDDAAGLQIA) originates from bacterial flagellin (*Felix et al., 1999*). Both peptides were synthesized (Genscript Inc) and reconstituted in $H_2O$. Leaflets were lightly scratch wounded with a fresh razor blade to remove cuticle area, and 10 μL of $H_2O$ with or without peptide (1 μM) was equally spread over the wounds with a pipette tip. After 1 hr, leaflets were excised and placed in sealed tubes for 2 hr before headspace sampling (1 mL). Ethylene was measured as previously described with a gas chromatographer (HP 5890 series 2, supelco #13,018 U, and 80/100 Hayesep Q 3 FT × 1/8IN x 2.1 MM nickel) with flame ionization detection and quantified using a standard curve (Scott, 99.5% ethylene, [Cat. No 25,881 U]; *Supplementary file 5b*; *Schmelz et al., 2003*). Subsequently, R (v4.0.3) and the R-packages dplyr (v1.0.7), ggpubr (v.0.4.0) and ggplot2 (v3.3.3) were used to analyze and plot the data; statistics were performed by using the paired Wilcoxon signed-rank test (*R Development Core Team, 2015*; *Wickham and Francois, 2015*; *Wickham, 2009*). The resulting figure was edited in Corel-DRAW Home & Student x7.

### ROS production in legumes
Leaf punches were taken with a 4 mm biopsy punch and floated in 150 μL of $H_2O$ using individual cells of a white 96-well white bottom plate (BRAND*plates* F pureGrade S white [REF 781665]). After overnight incubation, ROS production was measured upon addition of a 100 μL assay solution which contained 10 μg/mL luminol-horseradish peroxidase, 17 μg/mL luminol and the treatment (2.5 μM In11, 2.5 μM flg22 or $H_2O$). Luminescence was quantified with a TECAN SPARK plate reader every minute for an hour using an integration time of 500 ms. Four technical replicates were quantified for each treatment, and significant differences were determined by performing a two-group Mann-Whitney U Test between both treatments. R and the R-packages dplyr (v1.0.7), ggpubr (v.0.4.0), and ggplot2 (v3.3.3) were used to analyze and plot the data. The resulting figure was edited in Corel-DRAW Home & Student x7.

### Genome sequencing and assembly of legume species
Leaf tissue of legume plants was harvested and ground using an $N_2$-chilled mortar and pestle. In contrast to *P. erosus,* nuclei isolation was first performed on frozen tissue for *C. ensiformis*, *C. tetragonoloba*, and *H. podocarpum* using the Bionano Plant Tissue Homogenization Buffer (Part number 20283). Nuclear DNA was extracted for all above mentioned species with a modified CTAB(Cetyltrimethyl

ammonium bromide) protocol as described previously (*Lutz et al., 2011*). Resultant high-molecular weight (HMW) DNA concentrations were determined by Qubit and Bioanalyzer.

A modified protocol using the Oxford Nanopore Technologies (ONT) Rapid barcoding kit (SQK-RBK004) was used for sequencing. Briefly, 27 µL of ~20 ng/µL HMW DNA was combined with 3 µL of Rapid barcoding fragmentation mix and incubated for 1 min at 30°C followed by 1 min at 80°C. Ampure beads were added at a 0.7× final concentration and bead clean-ups performed as described in the Rapid barcoding kit (SQK-RBK004). All remaining steps were performed as described in the Rapid barcoding kit protocol. Each sample was sequenced on a single MinION or PromethION flow-cell (R9.4). High-accuracy base calling was performed in real time with MinKnow (v20.10.6).

Illumina short read sequence was generated for *C. tetragonoloba* and *H. podocarpum* from the same HMW DNA used for ONT long read sequencing. Paired end 2×150 bp sequence was generated on the Illumina NovaSeq6000 platform. In addition, we generated Illumina short reads for *C. ensiformis*, *P. erosus*, and *Macroptylium lathyroides* (Genewiz Inc). For these samples, genomic DNA was extracted using the Nucleospin Plant II kit (Macherey-Nagel). The Oxford Nanopore sequencing data and Illumina hiseq reads used in this study can be found in SRA(Short Read Archive) under Bioproject: PRJNA820752. The final genome assemblies are available on NCBI (*P. erosus*: JAMLFT000000000, *C. ensiformis*: JAMKYM00000000, *H. podocarpum*: JALJEV000000000, and *C. tetragonoloba*: JALJEW000000000).

The *P. erosus* genome was assembled using SPAdes (v3.15.4) using the meta option with both the Illumina and Oxford Nanopore readsets. Additional genome assemblies were produced with FlyE (v2.8.1), consensus was generated with three rounds of Racon (v1.3.1), and finally polished with Pilon (v1.22) three times with Illumina reads (2×150 bp). Final assembly quality was determined with assembly-stats and BUSCO (v5.3.0).

## Analysis of the contiguous *INR* locus and LRR-RLP homologs

The analyzed genome assemblies, versions, and their sources for the 20 legume species included in the contiguous *INR* locus analysis can be found in *Supplementary file 6a*; (*Lonardi et al., 2019*; *Kang et al., 2014*; *Kang et al., 2015*; *Schmutz et al., 2010*; *Chang et al., 2019*; *Varshney et al., 2011*; *Schmutz et al., 2014*; *Xie et al., 2019*). The nucleotide sequences of the extracted loci and their coordinates can be found respectively in *Supplementary file 1* and *Supplementary file 6a*. All INR and INR-like AA sequences included in the phylogenetic analysis can be found in *Supplementary file 2*. Newly sequenced and assembled genomes were analyzed to define the contiguous *INR* locus, *INR*, and *INR*-like sequences. Similarly, if not yet annotated in publicly available genomes, syntenic *INR* and *INR-like* homologs were identified. First, BLASTN (BLAST 2.9.0+, e-value 10) was used to identify the *INR* syntenic locus by mining the genomes for homologs of the strongly conserved neighbor (anchor) genes of common bean *INR* (*Phvul.007g077500*); *Phvul.007g077400* and *Phvul.007g077600*. Similarly, the genomes were mined for LRR-RLPs via BLASTN approach with a default e-value of 10, with exception of the *Glycine* pangenome where a BLASTP approach was used. The strongest LRR-RLP blast hits with *INR* were consistently identified in between the conserved neighbor genes. Finally, potential *INR* and *INR*-like ORFs(open reading frame) were determined by visual inspection in IGV (v2.10.3) and compared with the sequences of closely related annotated LRR-RLPs in other legume species. Additionally, *INR* homologs were identified in *M. lathyroides* by using an alternative method. First, *M. lathyroides* Illumina HiSeq paired end reads were mapped against both *Phvul.007G077500* (*INR*) and *PvUI111.07G078600* (*INR*-like) by using bwa (v0.7.17-r1188) (*Li and Durbin, 2009*). Second, mapped reads were sorted and indexed by using samtools (v 1.13; *Li et al., 2009*). Third, IGV (v2.10.3) was used to inspect the mapped reads and identify the SNPs of Mlathy *INR* and *INR*-like in comparison to the *P. vulgaris* homologs. Fourth, the above three steps were reiterated with the newly acquired gene sequences. Finally, the acquired Mlathy *INR* and *INR*-like sequences were confirmed by PCR using the Q5 Hot Start High-Fidelity kit (NEB), enzymatic clean-up using ExoSAP-IT (Thermo Fisher scientific), and Sanger sequencing of the entire amplified PCR product. Primers used in this process are listed in *Supplementary file 7a*.

## Contiguous *INR* locus analysis and validation of *P. erosus* receptor disruption

Locus comparison was performed using R (v4.0.3) and the R-package genoPlotR (v0.8.11) using the extracted contiguous *INR* loci and their corresponding annotation (*Supplementary file 1*). The resulting figure was edited in Corel-DRAW Home & Student x7. A PCR was performed to validate the disruption of the receptor at the syntenic locus of *P. erosus* using the Q5 Hot Start High-Fidelity kit (NEB). Primers used in the reaction are listed in *Supplementary file 7a*. Subsequently, the amplified PCR product was enzymatically cleaned-up using ExoSAP-IT (Thermo Fisher scientific). Finally, the disruption was validated by Sanger sequencing of the entire amplicon.

## Phylogenetic analysis of *INR* and *INR*-like homologs

Sequences were aligned using the online version of MAFFT 7 using the E-INS-i strategy (*Katoh et al., 2002*). One potential pseudogene of *H. podocarpum* was not incorporated in the phylogenetic analysis since the length of the sequence was about 79% of cowpea *INR* due to the absence of a part of the LRR domain in comparison to all other annotated receptors at the contiguous *INR* locus. A phylogenetic analysis was performed on the CIPRES web portal using RAXML-HPC2 on XSEDE (v8.2.12) with the automatic protein model assignment algorithm using ML criterion and 250 bootstrap replicates (*Miller et al., 2010*; *Stamatakis, 2014*). The DUMMY2 protein model was selected as the best scoring model for ML analysis. The resulting tree was rooted, visualized using MEGA10, and edited in Corel-DRAW Home & Student x7.

## NOTUNG analysis: prediction of duplication and gene loss events at the contiguous *INR* locus

First, a phylogenetic analysis was performed similar to the approach above with the sole difference that only *INR* and *INR*-like homologs located at the contiguous *INR* locus were included. Second, a species tree in Newick format was built which includes all species of which (1) *INR* and *INR*-like homologs were extracted, (2) *C. tetragonoloba* as it is the closest related legume species without a receptor at the contiguous *INR* locus, and (3) *M. truncatula* as the LRR-RLP used as an outgroup in the analysis was extracted from its genome. Third, the NOTUNG analysis was performed using the default options, including a duplication cost of 1.5 and a loss cost of 1 (v2.9.1.5; *Chen et al., 2000*). NOTUNG ignores incomplete lineage sorting as an evolutionary mechanism when both a rooted species and gene tree are used as input, as was the case for the present study.

## Molecular cloning of *INR* and *INR*-like homologs

Leaf tissue of legume plants was harvested and ground using an $N_2$-chilled mortar and pestle. Genomic DNA was extracted using the Nucleospin Plant II kit (Macherey-Nagel). RNA was extracted using the Nucleospin RNA Plant kit (Macherey-Nagel). cDNA was created using SuperScript IV Reverse Transcriptase (Invitrogen). All constructs were created using a hierarchical modular cloning approach facilitated by the MoClo toolkit and the MoClo Plant Parts kit (*Weber et al., 2011*; *Engler et al., 2014*). LRR-RLPs with no introns were PCR amplified from genomic DNA (Q5 Hot Start High-Fidelity, NEB), and all others (*Phvul.007g246600* and *Vigun07g039700*) were amplified from cDNA. All primers used for amplification are listed in *Supplementary file 7a*. All amplified PCR fragments were gel extracted using the PureLink Quick Gel Extraction Kit (Thermo Fisher scientific) and purified and concentrated using the Monarch PCR and DNA Cleanup Kit. Subsequently, the PCR fragments were ligated in an L-1 acceptor vector. A second digestion/ligation step was completed to ligate multiple parts together and complete the CDS while inserting it in an L0 acceptor vector. Throughout the previous steps, recognition sites for BsaI and/or BpiI were removed from the CDS. All constructs were validated by Sanger sequencing upon completion. Finally, all constructs were completed by adding the following MoClo modules; 35s Cauliflower Mosaic Virus +5'UTR Tobacco mosaic virus (pICH51266), GFP (*Aequorea victoria*; pICSL50008), and the OCS1 terminator (pICH41432) (*Engler et al., 2014*).

## *N. benthamiana* transient expression and western blotting

Constructs were electroporated into *Agrobacterium tumefaciens* strain GV3101 (pMP90 and pSOUP). Overnight cultures were resuspended in 150 μM acetosyringone in 10 mM 2-(*N*-morpholino) ethane-sulfonic acid (MES), pH 5.6, and 10 mM $MgCl_2$. After 3 hr of incubation at room temperature, *N.*

*benthamiana* leaves of 5-week-old plants were infiltrated at an optical density of 0.45 at 600 nm (OD$_{600}$).

Leaf punches were taken 48 hr after infiltration of the *N. benthamiana* leaves to validate the expression of the constructs in *N. benthamiana* by western blot. Ground, frozen tissue was homogenized in a 3× lamellae buffer (50 mM Tris-Cl pH 6.8, 6% SDS, 30% glycerol, 16% β-mercaptoethanol, and 0.006% bromophenol blue) and then cleared by centrifugation (10 m, 20,000 rcf). Subsequently, proteins in the supernatant were separated by performing an SDS-PAGE on an 8% acrylamide gel. Finally, a western blot was performed to visualize the GFP-tagged heterologously expressed proteins and actin as a loading control with respectively an α-GFP polyclonal (A-6455; Thermo) primary antibody at a 1:2000 dilution and an anti-Actin (ab197345 – A0480, abcam – Sigma-Aldrich) primary antibody at a 1:5000 dilution. α-rabbit (A6154; Sigma) was used as a secondary antibody for both at 1:10,000 dilution.

## ROS measurements in *N. benthamiana*

Following *Agrobacterium* infiltration for receptor expression (24 hr), leaf punches were taken with a 4 mm biopsy punch and floated in 150 µL of H$_2$O using individual cells of a white 96-well white bottom plate (BRAND*plates* F pureGrade S white). Subsequently, the same procedure was followed as outlined for ROS production in legumes. Four biological replicates were quantified (n=4 plants), with each biological replicate representing six technical replicates. R and the R-packages dplyr (v1.0.7), ggpubr (v.0.4.0), and ggplot2 (v3.3.3) were used to analyze and plot the data. The resulting figure was edited in Corel-DRAW Home & Student x7.

## Ethylene in *N. benthamiana*

For ethylene assays in *N. benthamiana*, a fully expanded leaf of 5-week-old plants was infiltrated with H$_2$O or 1 µM In11 with a blunt syringe. Subsequently, four leaf discs within the infiltrated area were immediately excised with a no. 5 cork borer and sealed in tubes (*Steinbrenner et al., 2020*). Headspace ethylene was measured after 3 hr of accumulation as described above. Subsequently, R and the R-packages dplyr (v1.0.7), ggpubr (v.0.4.0), and ggplot2 (v3.3.3) were used to plot the data and perform the statistics (paired Wilcoxon signed-rank test). The resulting figure was edited in Corel-DRAW Home & Student x7.

## Construction of chimeric receptors

Chimeric LRR-RLP constructs were generated using the MoClo toolkit (*Weber et al., 2011*). MoClo overhangs were designed in such a way that specific fragments amplified from the L0 constructs of *Vigun07g219600* and *Vigun07g219700* could be ligated in the preferred direction and order in an L0 vector. Primers used in these reactions are listed in *Supplementary file 7a*. All constructs were validated by Sanger sequencing upon completion. Subsequently, CDS stored in L0 universal acceptors was combined with the same MoClo modules as the earlier described LRR-RLP constructs mentioned above to create a complete L1 construct. Transient expression of chimeric receptor constructs in *N. benthamiana* was validated with western blot.

## ASR of LRR domain

The LRR (C1-3 domain, *Figure 4*) were extracted from all LRR-RLP sequences included in the earlier mentioned phylogenetic analysis (*Figure 3a*, *Supplementary file 4*). The LRR domain and subdomains of *Vu*INR were earlier identified using LRRfinder (*Steinbrenner et al., 2020*; *Offord et al., 2010*). Phylogenetic trees were built using MEGA X software (*Kumar et al., 2018*) and bootstrap method based on 1000 iterations. A codon-based 2172-nucleotide-long alignment was generated using MUSCLE (*Edgar, 2004*). NJ clustering method was used for constructing the codon-based tree on maximum composite likelihood substitution models. The ML tree was calculated using the GTR + G + I submodel as implemented in MEGA X software (*Kumar et al., 2018*). The resulting ML tree was used for the ASR of selected nodes of interest. Joint and marginal ASR were performed with FastML software using Jukes and Cantor substitution model for nucleotides, gamma distribution, and 90% probability cutoff (*Ashkenazy et al., 2012*). Finally, the sequences were domesticated to facilitate MoClo cloning by removing the BsaI and BpiI cut sites (*Supplementary file 4*).

## Construction of receptors with an ancestral reconstructed LRR domain

Ancestral reconstructed LRR-RLP constructs were generated using the MoClo toolkit (*Weber et al., 2011*). MoClo overhangs were designed in such a way that the AB domain and the DEFG domain amplified from *Vigun07g219600* could be ligated with the synthesized ancestral reconstructed LRR domain (C domain), resulting in an L0 vector. Primers used in these reactions are listed in *Supplementary file 7a*. All constructs were validated by Sanger sequencing upon completion. Subsequently, the complete CDS stored in L0 acceptors was combined with the same MoClo modules as the earlier described LRR-RLP constructs mentioned above to create a complete L1 construct. Transient expression of the ancestral reconstructed LRR-RLPs in *N. benthamiana* was validated with western blot.

## Construction of N4 ancestral receptor variants with a single AA substitution

N4 ancestral receptors with a single AA were created with two different approaches due to relatively low cloning efficiencies. N4 Y91F, N4 H239D, N4 A359S, and N4 H404N were created with the NEB Q5 Site-Directed Mutagenesis kit with as input the L1 vector of the N4 ancestral receptor. Primers used in these reactions are listed in *Supplementary file 7a*. The N4 R406K constructs was generated using the MoClo toolkit (*Weber et al., 2011*). MoClo overhangs were designed in such a way that they substituted the AA of interest to the corresponding AA of N3; the two fragments were amplified from the N4 ancestral construct, ligated in the MoClo L0 universal backbone, and subsequently ligated simultaneously with all other earlier described Moclo modules to complete the L1 construct. Primers used in these reactions are listed in *Supplementary file 7a*. All created constructs, independent of the cloning strategy, were validated by Sanger sequencing and transient expression was confirmed in *N. benthamiana* with western blot.

## AlphaFold prediction of *Vu*lNR

A protein structure prediction of Vigun07g219600 was created using the ColabFold platform (v1.3.0; *Mirdita et al., 2022*). The alphaFold2 input sequence alignment was generated through MMseqs2 using the unpaired+paired mode without using templates (*Jumper et al., 2021*; *Mirdita et al., 2019*; *Mirdita et al., 2017*; *Mitchell et al., 2020*). Three recycles were run for each of the five created models by AlphaFold2-ptm. The five resulting models were ranked based on the pLDDT score, and the highest scoring one was visualized using PyMOL (v2.5.2) (*PyMOL, 2020*; *Figure 6—figure supplement 2*).

## Acknowledgements

S.S. is supported as a Belgian American Educational Foundation postdoctoral fellow and the Mary Race Bevis Postdoc Research Award. S.S. and A.D.S. are supported by start-up funding from the University of Washington. A.D.S. is a Distinguished Investigator of the Washington Research Foundation.

## Additional information

### Funding

| Funder | Grant reference number | Author |
| --- | --- | --- |
| Belgian American Educational Foundation | | Simon Snoeck |
| Washington Research Foundation | | Adam D Steinbrenner |
| University of Washington | | Adam D Steinbrenner |

The funders had no role in study design, data collection and interpretation, or the decision to submit the work for publication.

## Author contributions
Simon Snoeck, Conceptualization, Data curation, Formal analysis, Funding acquisition, Investigation, Visualization, Methodology, Writing - original draft, Project administration; Bradley W Abramson, Data curation, Formal analysis, Investigation, Methodology; Anthony GK Garcia, Data curation, Investigation; Ashley N Egan, Supervision, Validation; Todd P Michael, Resources, Supervision; Adam D Steinbrenner, Conceptualization, Resources, Supervision, Funding acquisition, Methodology, Project administration, Writing – review and editing

## Author ORCIDs
Simon Snoeck ![ORCID] http://orcid.org/0000-0002-5288-0308
Anthony GK Garcia ![ORCID] http://orcid.org/0000-0003-4650-1348
Ashley N Egan ![ORCID] http://orcid.org/0000-0001-7803-4444
Adam D Steinbrenner ![ORCID] http://orcid.org/0000-0002-7493-678X

## Decision letter and Author response
Decision letter https://doi.org/10.7554/eLife.81050.sa1
Author response https://doi.org/10.7554/eLife.81050.sa2

# Additional files

## Supplementary files
• MDAR checklist

• Supplementary file 1. Fasta file of nt sequences of the inceptin receptor (INR) syntenic loci incorporated in the contiguous INR locus analysis.

• Supplementary file 2. Fasta file of the amino acid (AA) sequences inceptin receptor (INR) and INR-like homologs included in the phylogenetic analysis.

• Supplementary file 3. Fasta file containing the sequences of the chimeric receptors.

• Supplementary file 4. Fasta file containing the nucleotide leucine-rich repeat (LRR) domain sequences of the inceptin receptor (INR) and INR-like homologs used for the ancestral sequence reconstruction (ASR) analysis and the resulting predicted (and domesticated for MoClo) LRR domain sequences for the ancestral nodes of interest.

• Supplementary file 5. Excel file containing (A) overview of the used legume species, accessions, and their origin, and (B) overview of ethylene response data of legume species.

• Supplementary file 6. Excel file containing (A) overview of mined assemblies of contiguous inceptin receptor (INR) loci (+ coordinates) and leucine-rich repeat (LRR) receptor-like proteins (RLPs), (B) genome assembly stats, and (C) output file of CENSOR analysis which detected repetitive elements of the INR loci by comparison to known repeats of the *Arabidopsis thaliana* Repbase database.

• Supplementary file 7. Excel file containing (A) primers used in this study, and (B) p-values of statistical analysis depicted in all figures and figure supplements.

• Supplementary file 8. Amino acid (AA) sequence alignment of the leucine-rich repeat (LRR) domains of the inceptin receptor (INR) clade, INR-like 1, clade and the closest ancestral reconstructed nodes with differential In11-response. Differences or consensus (dashed marks) from ancestral sequence reconstruction (ASR) construct N3 are indicated for all other sequences. The five AA polymorphisms between N3 and N4 that are similar within all LRR-RLPs of the INR clade and different in LRR-RLPs of the INR-like 1 sister clade are highlighted in yellow. C1-domain starts at the N-terminus, C3-domain ends at the C-terminus, C1 and C3 domain are separated by the C2-domain which is highlighted in the frame.

## Data availability
Sequencing data and genome assemblies have been deposited on NCBI under the following bioprojects: PRJNA817236, PRJNA817235, PRJNA817241, PRJNA817237 and PRJNA817234. Figure 1 - Suppl. data 2 contains the numerical data used to generate this figure.

The following datasets were generated:

| Author(s) | Year | Dataset title | Dataset URL | Database and Identifier |
|---|---|---|---|---|
| Snoeck S | 2022 | Hylodesmum podocarpum, de novo genome assembly | https://www.ncbi.nlm.nih.gov/bioproject/?term=PRJNA817236 | NCBI BioProject, PRJNA817236 |
| Snoeck S | 2022 | Canavalia ensiformis, de novo genome assembly | https://www.ncbi.nlm.nih.gov/bioproject/?term=PRJNA817235 | NCBI BioProject, PRJNA817235 |
| Snoeck S | 2022 | Macroptilium lathyroides Raw sequence reads | https://www.ncbi.nlm.nih.gov/bioproject/?term=PRJNA817241 | NCBI BioProject, PRJNA817241 |
| Snoeck S | 2022 | Cyamopsis tetragonoloba, de novo genome assembly | https://www.ncbi.nlm.nih.gov/bioproject/?term=PRJNA817237 | NCBI BioProject, PRJNA817237 |
| Snoeck S | 2022 | Pachyrhizus erosus, de novo genome assembly | https://www.ncbi.nlm.nih.gov/bioproject/?term=PRJNA817234 | NCBI BioProject, PRJNA817234 |

The following previously published datasets were used:

| Author(s) | Year | Dataset title | Dataset URL | Database and Identifier |
|---|---|---|---|---|
| Hane JK | 2016 | Lupinus angustifolius Genome sequencing and assembly | https://www.ncbi.nlm.nih.gov/bioproject/299755 | NCBI BioProject, 299755 |
| Varshney RK | 2013 | Cicer arietinum Genome sequencing and assembly | https://www.ncbi.nlm.nih.gov/bioproject/?term=PRJNA175619 | NCBI BioProject, PRJNA175619 |
| Tang H | 2014 | Medicago truncatula strain:A17 Genome sequencing and assembly | https://www.ncbi.nlm.nih.gov/bioproject/?term=PRJNA10791 | NCBI BioProject, PRJNA10791 |
| De Vega JJ | 2015 | Assembly of red clover (Trifolium pratense L.) genome | https://www.ncbi.nlm.nih.gov/bioproject/?term=PRJEB9186 | NCBI BioProject, PRJEB9186 |
| Los Alamos National Laboratory | 2018 | Abrus precatorius isolate:BTA Genome sequencing and assembly | https://www.ncbi.nlm.nih.gov/bioproject/482671 | NCBI BioProject, 482671 |
| African Centre of excellence in Phytomedicine Resaerch | 2018 | Mucuna pruriens isolate:JCA_2017 Genome sequencing and assembly | https://www.ncbi.nlm.nih.gov/bioproject/?term=PRJNA414658 | NCBI BioProject, PRJNA414658 |
| Varshney RK | 2011 | Cajanus cajan Genome Sequencing | https://www.ncbi.nlm.nih.gov/bioproject/72815 | NCBI BioProject, 72815 |
| Xie M | 2019 | Genome sequencing and de novo assembly of Glycine soja W05 | https://www.ncbi.nlm.nih.gov/bioproject/486704 | NCBI BioProject, 486704 |
| Chang S | 2013 | Soybean WGS sequencing project | https://www.ncbi.nlm.nih.gov/bioproject/?term=PRJNA19861 | NCBI BioProject, PRJNA19861 |
| University of California, Riverside | 2019 | Vigna unguiculata isolate:L. Walp \| cultivar:IT97K-499-35 Genome sequencing and assembly | https://www.ncbi.nlm.nih.gov/bioproject/381312 | NCBI BioProject, 381312 |
| Yang K | 2015 | Vigna angularis cultivar:Jingnong 6 Genome sequencing and assembly | https://www.ncbi.nlm.nih.gov/bioproject/261643 | NCBI BioProject, 261643 |

*Continued*

| Author(s) | Year | Dataset title | Dataset URL | Database and Identifier |
|---|---|---|---|---|
| Kang YJ | 2014 | Vigna radiata var. radiata cultivar:VC1973A Genome sequencing and assembly | https://www.ncbi.nlm.nih.gov/bioproject/243847 | NCBI BioProject, 243847 |
| Chang Y | 2019 | Genome sequencing of four African orphan crops | https://www.ncbi.nlm.nih.gov/bioproject/?term=PRJNA474418 | NCBI BioProject, PRJNA474418 |
| Andes UDL, Colombia UND | 2020 | Phaseolus lunatus genomic resources | https://www.ncbi.nlm.nih.gov/bioproject/?term=PRJNA596114 | NCBI BioProject, PRJNA596114 |
| Michigan StateUniversity | 2020 | Genome and transcriptome sequencing of Phaseolus acutifolius and Phaseolus vulgaris | https://www.ncbi.nlm.nih.gov/bioproject/?term=PRJNA607288 | NCBI BioProject, PRJNA607288 |
| JGI-PGF JGI | 2013 | Phaseolus vulgaris cultivar:G19833 Genome sequencing and assembly | https://www.ncbi.nlm.nih.gov/bioproject/?term=PRJNA41439 | NCBI BioProject, PRJNA41439 |

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
