## [Editor Report]

This manuscript, of interest to those studying the evolution of immunity, investigates the evolutionary history of a recently described herbivore-associated molecular pattern (HAMP) receptor, INR, which perceives the caterpillar-derived peptide HAMP, In11. The authors compare INR homologs to identify evolutionarily conserved residues and use chimeric fusion proteins to investigate specificity. The findings presented are valuable and supported by convincing experiments and analysis.

---

## [Decision Letter]

**Decision letter after peer review:**

Thank you for submitting your article "Evolutionary gain and loss of a plant pattern-recognition receptor for HAMP recognition" for consideration by *eLife*. Your article has been reviewed by 2 peer reviewers, and the evaluation has been overseen by a Reviewing Editor (me!) and Detlef Weigel as the Senior Editor. The following individual involved in the review of your submission has agreed to reveal their identity: Daniil M Prigozhin (Reviewer #1).

We have discussed your manuscript. Overall, we are all very positive about your work. The paper is nicely constructed and was fun to read. However, both reviewers would like to see a bit more analysis/interpretation of the predicted structure and I had one technical concern. Please see below for detail. I have drafted the following to help you prepare a revised submission. Please feel free to reach out should you require any clarification.

Essential revisions:

1) Please include additional analysis/interpretation based on structural predictions (see detailed comments from both reviewers, below). In addition, this may allow additional interpretation of the data including the auto activity of 219600-C1.

2) Both reviewers made several suggestions for ways to improve clarity and correct minor typos.

3) I had one small technical question – I notice that the ROS responses are cyclic for some receptors and a single burst for others. Any thoughts on this? Also, in a few cases, you see a response with ethylene but not ROS or the other way around. My guess is that these are false positives. Do you agree? For example Glyma_10G228000. Perhaps a sentence or two in the discussion is warranted.

*Reviewer #1 (Recommendations for the authors):*

1. Independent losses of INR are important to the overall understanding of the story. Following up on the likely genomic events that lead to several independent losses is advisable.

2. Evolution of protein receptor recognition is best interpreted in the context of protein structure. It is up to the authors whether this makes main figures or not, but including a predicted protein model with N3 to N4 and N3 to N14 changes highlighted should at least make a supplemental figure. Alphafold/Rosettafold results are well accepted as a best guess estimates absent an experimental structure.

3. For the ROS time courses, unifying the Y-axis across figures would improve sample-to-sample comparison. Same for the X-axis of the ethylene subfigures of Figure 3 to Figure 5.

4. The use of "autoactivity" (in line 374) with regards to the mutant 219600-c1 is surprising, and the nature of the differences between 219600-F, C3, C2, and C1 is somewhat underexplored. Would the 219600-c1 receptor signal in the absence of scratch wounding etc? Is there a phenotype in the transient assay after protein expression but before any other manipulations are performed? I would agree with a "responded to control treatment only" characterization absent more details.

(I believe C2 is likely critical for relaying binding at C1 down to the co-receptor, so a mismatch between C1 and C2 is bound to be "strange". The differences between +/- peptides could be due to the pH of the peptide solution, its reduction potential, or some other general property. I also found the pulse shape difference between -C2 and -F absolutely fascinating, yet not mentioned in the paper.)

*Reviewer #2 (Recommendations for the authors):*

Line 44 – include Benthamiana RLPs – several have been identified here e.g. RXEG1 and RE02.

Whilst LRR-RLP structures have not currently been solved, based upon what we know from other LRR domains predictions of structure should be quite accurate. It would be nice to highlight the residues that you identify as being important for the recognition of a structural prediction. This could help the reader.

Figure 1 – it could be interesting to show the correlation between In11-induced ethylene and flg22-induced ethylene (Sup Figure 1 a). Is there a strong correlation?

158 – 'contained a orthologs'.

Figure 4 b – why does mock give a response in 219600? Would you expect a burst if it is constitutive autoactivity as suggested (374)? Does this RLP induce cell death? Is the burst in the mock reproducible? Also, the burst seems weaker following treatment in the rep shown. If it was autoactive I think it would not matter whether it is mock or treatment.

For expression western blots it would have been nice to include a -ve control of an untransformed leaf – however, this is not strictly required as some of the receptors have slightly different sizes.

Whilst perception seems to have been lost in G. max/G. soja it would be interesting to know how widely INR is present in the pangenome of a responsive species. In Arabidopsis, it seems that many RLPs have significant presence/absence polymorphism. It would be interesting to know if this is the case with INR, however, I understand that the availability of genomic/germplasm resources may be limiting. E.g. what proportion of cowpea accessions are In11 responsive?

Whilst many of the receptors are not responsive to In11, they could also be functional receptors for an unknown ligand. I think this could be made more apparent in the text. One option could be polymorphisms within In11, is it known if these exist? E.g. 327-328.

Does INR1-like form a constitutive interaction with SOBIR1? I would expect it can be given the activity of the C2 chimera, however, it could be nice to demonstrate this via Co-IP.

It would be nice to test each of the 5 substitutions between N3 and N4 for a loss of In11 perception in Benth, however, I do not think this is necessary.

316 – 'unusually long evolutionary history' – please provide more context what are you comparing with? It would be nice to add context to what is currently known.

Personally, I think the discussion could be made more concise.

326 – I would not agree that it represents an 'alternative' mechanism – I don't think there is currently a paradigm for how these receptors evolve. I think this work represents one of the few attempts to characterise the origin of the receptor in detail.

349 – gives the impression that EFR/XPS1/CORE is RLPs – rephrase. I would also argue that for some of the receptors there has been some extensive phenotyping than others – e.g. for RLP42.

356 – I would argue that this paper suggests that RLPs have dynamic evolutionary patterns – similar to NLRs, unlike RKs (in general). They highlight two examples that are conserved, but I would not say that these examples are representative.

I would prefer that P-values are also shown on the graphs (potentially in addition to the notation of significance).

I am not qualified to comment on the quality of the sequencing – but I believe it represents a valuable resource of the community and is sufficient to support the claims in the manuscript.

364 – I would consider TMM a ligand binding LRR-RLP (https://www.ncbi.nlm.nih.gov/pmc/articles/PMC5458759/) although I appreciate it is significantly

I would recommend citing this paper for the lack of RLP structure https://www.ncbi.nlm.nih.gov/pmc/articles/PMC6503760/.

---

## [Author Response]

Essential revisions:1) Please include additional analysis/interpretation based on structural predictions (see detailed comments from both reviewers, below). In addition, this may allow additional interpretation of the data including the auto activity of 219600-C1.

We have generated an AlphaFold-based structural prediction of INR. In addition, we have included clarified interpretations of the 219600-C1 construct, which showed ROS response to water-only treatment. Please see specific responses to reviewers for additional details.

2) Both reviewers made several suggestions for ways to improve clarity and correct minor typos.

We appreciate the suggestions and corrections of both reviewers and incorporated them in the manuscript.

3) I had one small technical question – I notice that the ROS responses are cyclic for some receptors and a single burst for others. Any thoughts on this? Also, in a few cases, you see a response with ethylene but not ROS or the other way around. My guess is that these are false positives. Do you agree? For example Glyma_10G228000. Perhaps a sentence or two in the discussion is warranted.

We are also interested in cyclic ROS responses but unable at this stage to account for differences. We tested if cyclic ROS was a function of the C-terminal GFP tag. Although more subtle, cyclic ROS responses were still observed when two receptor homologs did not have a C-terminal GFP tag. Both constructs also gave equivalent ethylene burst to the original GFP tagged constructs, so it is not clear that cyclic ROS translates into different downstream response. We have added this data on the effect of C-terminal tag in a new Figure 3—figure supplement 2. Our overall model is that each ortholog or tagged variant interfaces slightly differently with *N. benthamiana* signaling machinery to activate RBOHD. We added the following sentences to address the observation in the results:

“Intriguingly, cyclic ROS responses were observed for three out of five In-11 responding receptors (Vigun07g219600, Phvul.007G077500 and Mlathy INR). Corresponding constructs without a C-terminal GFP tag reduced cyclic characteristics of the ROS response but resulted in equivalent ethylene bursts (Figure 3—figure supplement 2).”

Regarding ROS versus ethylene correspondence -- ROS responses were previously shown with scaled y-axes, which inflated the apparent size of certain responses. This is now adjusted to only use one of 3 scales throughout all figures. In11-induced ROS and ethylene responses were complementary for all constructs. Nevertheless, the results for Glyma_10G228000 are indeed remarkable as our control treatment resulted in a low and late ROS burst relative to any of the other ROS bursts observed in this study for other constructs. Similarly, the control treatment resulted in a higher ethylene production for this construct.

Reviewer #1 (Recommendations for the authors):1. Independent losses of INR are important to the overall understanding of the story. Following up on the likely genomic events that lead to several independent losses is advisable.

We agree that the INR loss events, potentially mediated via TE activity, are intriguing and of interest for further investigation. We therefore used the CENSOR tool to annotate repetitive elements in INR loci using the Repbase database with Arabidopsis as a sequence source.

In the INR locus of jicama (*Pachyrhizus erosus*), multiple fragments of an LRR-RLP receptor can be found by BLAST analysis as indicated by Figure 2b. Transposon annotation with CENSOR revealed a hit with a partial transposable element of the DNA/Mariner class (117 bp) in between two LRR-RLP fragments, possibly consistent with an ancient full transposition interrupting an LRR-RLP and/or INR function (Figure 2—figure supplement 1, supplementary table 5). We added the following sentence to the results:

“A Mariner-like transposase element exists between the LRR-RLP gene fragments (Figure 2—figure supplement 1, supplementary table 5).”

Beyond the example in jicama, we found various TE hits in all INR loci of the Millettioids. This finding aligns well with structural variation and variable number of LRR-RLPs found at the corresponding INR loci (Figure 2 and Suppl. Figure 2). We adapted the result section as follows:

“The organization of the INR locus is highly diverse among legumes, with zero to seven LRR-RLP encoding-genes per species in addition to extensive repetitive sequence and transposable element content (Figure 2a, Figure 2-supplement figure 1, supplementary table 5).”

And the discussion as follows:

“In contrast, *P. vulgaris* and *M. pruriens* lost INR-like, and *P. erosus* seems to have lost both INR and INR-like as it does not contain any complete coding sequence of an LRR-RLP at the INR locus. Repetitive elements and transposable elements within the INR locus may have disrupted ancestral INR and INR-like genes in these lineages, and can be found between LRR-RLP genes and gene fragments (Figure 2-supplement figure 2). Loss of INR (and INR-like) may reflect the propensity of tandemly duplicated loci to lose functions through gene conversion, and/or reciprocal gene loss (62).”

While intriguing, we believe we cannot draw firm conclusions about the role of these elements with respect to the loss of INR in specific species. For example, high-quality genomes of close Glycininae outgroups would be needed to identify specific losses of *Glycine max* and tight correlation with TE variation*.* Nevertheless, we agree that this could be an interesting and achievable project in the near future upon release of additional legume genomes.

We are also hesitant to draw stronger conclusions due to the TE source used for our CENSOR analysis. Genome specific pipelines for TE annotation would improve the analysis as the use of a TE database from a distantly related species (Arabidopsis) decreases annotation quality (Platt 2016, https://doi.org/10.1093/gbe/evw009). Manual curation of automated predictions is also advisable as discussed by (Goubert *et.al.* 2022, https://doi.org/10.1186/s13100-021-00259-7). Fine-scale tracking of TE behavior within the Glycininae would allow genome-wide analysis of both NLR and RLP evolution.

2. Evolution of protein receptor recognition is best interpreted in the context of protein structure. It is up to the authors whether this makes main figures or not, but including a predicted protein model with N3 to N4 and N3 to N14 changes highlighted should at least make a supplemental figure. Alphafold/Rosettafold results are well accepted as a best guess estimates absent an experimental structure.

We thank the reviewer for this suggestion. We now include a new main figure (Figure 6) which includes a structural prediction of INR (Vigun07g219600). In this structure we highlight 5 AA of interest based on analysis of N4 vs N3, the respective ancestors of the INR clade and the combined INR+INR-like clade. We also added additional functional data which shows that H404 and R406 are essential for function in the N4 ancestrally reconstructed construct. For more details, please see the adapted manuscript (305-336, 479-490 and 705-723).

3. For the ROS time courses, unifying the Y-axis across figures would improve sample-to-sample comparison. Same for the X-axis of the ethylene subfigures of Figure 3 to Figure 5.

We thank the reviewer for this suggestion. We agree that a consistent use of x- and y-axes limits improves the figure readability. We now use three y-axes for ROS with small changes in upper limit for readability (0-2000, 0-4000, or 0-6000 RLU) rather than previous mixed axes with scales as fine as 0-400 RLU. We changed the main Figures 3-5 and the newly added figures (Figure 6 and Figure 3—figure supplement 2) to the manuscript accordingly for both ROS and ethylene data.

4. The use of "autoactivity" (in line 374) with regards to the mutant 219600-c1 is surprising, and the nature of the differences between 219600-F, C3, C2, and C1 is somewhat underexplored. Would the 219600-c1 receptor signal in the absence of scratch wounding etc? Is there a phenotype in the transient assay after protein expression but before any other manipulations are performed? I would agree with a "responded to control treatment only" characterization absent more details.

We agree with the reviewer and changed the text accordingly (new text underlined):

“(Line 244) Intriguingly, the chimeric receptor 219600−C1, containing C1 of VuINR and C2 of VuINR-like, responded to the control treatment.”

(Line 445) “Notably, a chimeric receptor with mixed C1 and C2, 219600-C1, is the sole construct that responded to the control treatment, suggestive of a critical role for C1-C2 compatibility.”

Luminescence for the ROS assays is measured in leaf disk punches. Hence, we are unable to measure ROS without creating leaf damage which potentially has an influence on the observed ROS response to control treatment. We attempted to further explore the heightened responsiveness of 219600-C1 using two additional approaches. First, we expressed chimeric receptor 219600-C1 and parent receptors without any subsequent manipulations to check for a potential phenotype by eye, such as cell death or chlorosis. No phenotype was observed 7 days after infiltration. We show this lack of response in a new panel, Figure 4—figure supplement 2a. Secondly, we hypothesized that heightened ROS responses to control treatment might manifest as cell death responses to PAMPs or HAMPs. We therefore repeated the above-described set-up except that 24h post infiltrations, we infiltrated the following treatments: H_2_O, 1 µM In11 and 1 µM flg22. Similarly, no phenotypes were observed for 7 days after construct infiltration (Figure 4—figure supplement 2b). We added the following sentences to the results:

“Intriguingly, the chimeric receptor 219600−C1, containing C1 of VuINR and C2 of VuINR-like, responded to the control treatment. Nonetheless, no visible phenotype was observed seven days post infiltration of 219600-C1 in *N. benthamiana* with or without subsequent infiltration of flg22, In11 or H_2_O one day post the construct infiltration (Figure 4—figure supplement 2a-2b).”

(I believe C2 is likely critical for relaying binding at C1 down to the co-receptor, so a mismatch between C1 and C2 is bound to be "strange". The differences between +/- peptides could be due to the pH of the peptide solution, its reduction potential, or some other general property. I also found the pulse shape difference between -C2 and -F absolutely fascinating, yet not mentioned in the paper.)

We agree that multiple factors could result in what we observed and thank reviewer 1 for suggesting other hypotheses. We changed the text accordingly:

“Notably, a chimeric receptor with mixed C1 and C2, 219600-C1, is the sole construct that responded to the control treatment, suggestive of a critical role for C1-C2 compatibility.”

The interface between C1 and C2 is also informed by structural modeling. The AlphaFold prediction of INR (new Figure 6) suggests an interaction between C2 and the N-terminal part of the C1 LRR. Additionally, one of the five conserved AA (Y91) identified by our ASR analysis is predicted to play a part in this interaction. Y91F did not completely abolish ROS-burst although it was relatively low. We now note this C1-C2 predicted interaction in the Results section (lines 314-317).

Finally, we discussed in our response to the editor above that the cyclic ROS response seems to be somewhat dependent on the C-terminal tag and specific homologue used (new Figure 3—figure supplement 2). We therefore do not want to overinterpret the single peak of 219600-C2. However, we have added the following sentence acknowledging cyclic responses and the role of C-terminal tags.

“Intriguingly, cyclic ROS responses were observed for three out of five In-11 responding receptors (Vigun07g219600, Phvul.007G077500 and Mlathy INR). Corresponding constructs without a C-terminal GFP tag resulted in equivalent ethylene bursts, but ROS bursts only had relatively subtle cyclic characteristics (Figure 3—figure supplement 2).”

Reviewer #2 (Recommendations for the authors):Line 44 – include Benthamiana RLPs – several have been identified here e.g. RXEG1 and RE02.

We included a few extra examples of LRR-RLPs with identified molecular patterns for both *Nicotiana benthamiana* and *Brassica napus.* The introduction changed as follows (new text underlined):

“Evidence is accumulating for highly specialized roles of plant LRR-RLPs as immune sensors (15,18). Various LRR-RLPs from Arabidopsis, tomato, wild tobacco, rapeseed and cowpea have been shown to detect molecular patterns from fungi, bacteria, parasitic weeds and herbivores (3,15). For example, Cf-9, Cf-4 and Cf-2 interact with respective molecular patterns of *Cladosporium fulvum*, Avr9, Avr4 and Avr2 (via host protein Rcr3), and are restricted to the *Solanum* genus (19–25). Arabidopsis RLP42 recognizes fungal endopolygalacturonases (PG) eptitope pg9(At) derived from *Botrytis cinerea* (26,27). Similarly, RLP23 is an Arabidopsis-specific LRR-RLP, which recognizes nlp20 peptide from the NECROSIS AND ETHYLENE-INDUCING PEPTIDE1 (NEP1)-LIKE PROTEINS (NLPs) found in bacterial/fungal/oomycete species (15,28). RXEG1 and RE02 were identified in wild tobacco and are respectively triggered by the fungal elicitors XEG1 (*Phytophthora sojae*) and E02 (*Valsa mali*) (29,30). LepR3 (AvrLm1) and RLM2 (AvrLm2) convey both race specific resistance to the fungal pathogen *Leptosphaeria maculans* in rapeseed (31,32). ReMAX is restricted to the Brassicaceae and triggered by the MAMP eMax originating from *Xanthomonas* (33). Cuscuta Receptor1 (*CuRe1*) is specific to *Solanum lycopersicum* and senses the peptide Crip21 which originates from parasitic plants of the genus *Cuscuta* (34,35)*.* Finally, the inceptin receptor (INR) appears to be specific to the legume tribe Phaseoleae and recognizes inceptin (In11), a HAMP found in the oral secretions of multiple caterpillars (36–39). Notably, all the above examples of LRR-RLPs are family-specific, restricted to the Solanaceae (Cf-2, Cf-4, Cf-9, RXEG1, RE02 and CuRe1), Brassicaceae (RLP23, RLP42, LepR3, RLM2 and ReMAX) or Leguminosae (tribe Phaseoleae) (INR) (18,27). However, despite clear signatures of lineage-specific functions, specific evolutionary steps leading to novel LRR-RLP functions across multiple species have not been described.”

Whilst LRR-RLP structures have not currently been solved, based upon what we know from other LRR domains predictions of structure should be quite accurate. It would be nice to highlight the residues that you identify as being important for the recognition of a structural prediction. This could help the reader.

We thank the reviewer for the suggestion which was also given by reviewer 1. We agree and our response to reviewer 1 is reproduced here:

We now include a new main figure (Figure 6) which includes a structural prediction of INR (Vigun07g219600). In this structure we highlight 5 AA of interest from analysis of N4 vs N3, the respective ancestors of the INR clade and the combined INR+INR-like clade. We also added additional functional data which shows that H404 and R406 are essential for function in the N4 ancestrally reconstructed construct. For more details, please see the adapted manuscript (305-336, 479-490 and 705-723).

Figure 1 – it could be interesting to show the correlation between In11-induced ethylene and flg22-induced ethylene (Sup Figure 1 a). Is there a strong correlation?

We thank the reviewer for this interesting idea. Of 12 total In11-responding species, 4 species (*V. radiata*, *P. vulgaris*, *M. pruriens* and *L. purpureus*) do not respond to flg22 with an ethylene burst (Figure 1 and Suppl. Figure 1). Hence, ethylene responses to both elicitors are uncorrelated for those four species. Nevertheless, all four of them do respond to flg22 with a ROS burst.

In contrast, 8/12 species respond to both flg22 and In11 with an ethylene burst (*D. unicatum*, *C. cajan*, *T. labialis*, *M. lathyroides*, *R. minima*, *C. angustifolia*, *M. artropurpureum* and *V. unguiculata*). However, a strong correlation between the ethylene ratio of response against both elicitors is absent for these species as can be seen in Author response image 1.

**Author response image 1. sa2fig1:** 

158 – 'contained a orthologs'.

Corrected.

Figure 4 b – why does mock give a response in 219600? Would you expect a burst if it is constitutive autoactivity as suggested (374)? Does this RLP induce cell death? Is the burst in the mock reproducible? Also, the burst seems weaker following treatment in the rep shown. If it was autoactive I think it would not matter whether it is mock or treatment.

We do agree with reviewer 2 and agree that autoactivity is not the correct way to describe the behavior of this specific construct (219600-C1). We changed the text accordingly as suggested by reviewer 1 in both the result and the Discussion section:

“Intriguingly, the chimeric receptor 219600−C1, containing C1 of VuINR and C2 of VuINR-like, responded to the control treatment.”

The ROS burst following the mock treatment was consistent over multiple biological replicates measured on different days (i.e. different plant batches) and was not observed for any other constructs tested on the same leaves.

Reviewer 1 raised similar questions regarding 219600-C1 and we reproduce our response here:

Luminescence for the ROS assays is measured in leaf disk punches. Hence, we are unable to measure ROS without creating leaf damage which potentially has an influence on the observed ROS response to control treatment. We attempted to further explore the heightened responsiveness of 219600-C1 using two additional approaches. First, we expressed chimeric receptor 219600-C1 and parent receptors without any subsequent manipulations to check for a potential phenotype by eye, such as cell death or chlorosis (Figure 4—figure supplement 2a). No phenotype was observed 7 days after infiltration. Secondly, we hypothesized that heightened ROS responses to control treatment might manifest as cell death responses to PAMPs or HAMPs. We therefore repeated the above-described set-up except that 24h post infiltrations, we infiltrated the following treatments: H_2_O, 1 µM In11 and 1 µM flg22. Similarly, no phenotypes were observed for 7 days after construct infiltration (Figure 4—figure supplement 2b). We added the following sentences to the results:

“Intriguingly, the chimeric receptor 219600−C1, containing C1 of VuINR and C2 of VuINR-like, responded to the control treatment. Nonetheless, no visible phenotype was observed seven days post infiltration of 219600-C1 in *N. benthamiana* with or without subsequent infiltration of flg22, In11 or H_2_O one day post the construct infiltration (Figure 4—figure supplement 2a-2b).”

For expression western blots it would have been nice to include a -ve control of an untransformed leaf – however, this is not strictly required as some of the receptors have slightly different sizes.

Our revised manuscript includes an additional western blot for mutated ancestral sequence reconstructed (ASR) receptors. This time, we included an infiltration media only control (Figure 6, figure supplement 1).

Whilst perception seems to have been lost in G. max/G. soja it would be interesting to know how widely INR is present in the pangenome of a responsive species. In Arabidopsis, it seems that many RLPs have significant presence/absence polymorphism. It would be interesting to know if this is the case with INR, however, I understand that the availability of genomic/germplasm resources may be limiting. E.g. what proportion of cowpea accessions are In11 responsive?

We agree that the degree of conservation of INR is a very interesting aspect of its evolution. In a previous study, all 364 tested *Vigna unguiculata* accessions were In11 responsive (https://doi.org/10.1073/pnas.2018415117 – Table S2). Additional *Phaseolus vulgaris* and *Vigna unguiculata* reference genomes are now available on Phytozome, and all encode INR. We added the following sentence to the manuscript:

“Each of 364 tested varieties of *V. unguiculata* are able to respond to In11 and our analysis now extends this conservation across legume species (36).”

The recently published soybean pangenome (https://doi.org/10.1016/j.cell.2020.05.023) provided 26 de novo assemblies for *Glycine max and G. soja.* None have INR. We note this observation in the results and the discussion:

“To confirm that INR was lost in *Glycine* and not just in reference assemblies, we performed BLASTP searches using Vigun07g219700 AA sequence against 26 *G. soja* and *G. max* de novo assemblies from a recent pangenome analysis (56).”

“Two reference genomes as well as 26 de novo assemblies of the closely related Glycininae, *G. soya* and *G. max*, do not encode INR (56).”

Whilst many of the receptors are not responsive to In11, they could also be functional receptors for an unknown ligand. I think this could be made more apparent in the text. One option could be polymorphisms within In11, is it known if these exist? E.g. 327-328.

We agree with reviewer 2 and assume that they are functional receptors as they seem to be similarly conserved in comparison the In11-responsive INR-clade. However, so far, we can only hypothesize about the ligand it interacts with. Several polymorphisms do exist for In11 as shown in Author response image 2, which is based on the γ-subunit ATP synthase sequence from 15 legume species. However, we believe they most likely do not influence the interaction with INR based on the below arguments:

First, Schmelz 2006 (Figure 3B, https://www.pnas.org/doi/full/10.1073/pnas.0602328103#s-1) showed that *V. unguiculata* also responds quantitatively similar to the In11 variant from the monocot *Zea mays* (ICDVNGVCVDA), the exact same In11 variant which we identified in multiple Millettioids as shown in the attached alignment (*L. purpureus*, *G. max*, *G. soja*, *P. erosus*, *H. podocarpum* and *M. pruriens*).Second, Schmelz 2007 (Figure 8., https://doi.org/10.1104/pp.107.097154) investigated the effect of alanine substitutions on the elicitor potential of In11. Only mutations of positions 3, 8, and 10 of the mature In11 peptide (D314A, C319A and D321A) lost elicitation activity, and these are different from the polymorphic sites 4, 7, and 9 across legumes.

We have added a sentence to add our thinking about the function of INR-like:

“Certain Phaseoloids and earlier diverged non-Phaseoloids also contain INR-like homologues at the contiguous INR locus; thus, INR most likely arose from an existing LRR-RLP, ~28 mya ago. The role of INR-like homologues is not clear, but they may detect a related, In11-like ligand.”

Does INR1-like form a constitutive interaction with SOBIR1? I would expect it can be given the activity of the C2 chimera, however, it could be nice to demonstrate this via Co-IP.

We thank the reviewer for this idea and interest in INR1-like signaling mechanisms. As described in Steinbrenner 2020, *Vigna unguiculata* INR associates with both AtSOBIR1 and VuSOBIR1 (Vigun09g096400). However, no SOBIR1 association data is available for *Vigna unguiculata* INR-like1. We agree with reviewer 2 that this would be a valuable experiment and believe that Co-IP could potentially give us valuable insights not only for INR-like 1 but also for multiple chimeric and even ancestrally reconstructed constructs. However, we believe these experiments would be best performed in a full study of SOBIR1 association, BAK1/SERK association, and ligand binding – especially if a ligand for INR-like1 can be identified.

It would be nice to test each of the 5 substitutions between N3 and N4 for a loss of In11 perception in Benth, however, I do not think this is necessary.

We thank the reviewer for the interest in these sites, which we agree are interesting with AlphaFold structural context. We tested these mutations and now place data alongside the INR structural model. We added an extra main figure (Figure 6) which provides ROS and ethylene data of N4 constructs with the five single AA switches.

316 – 'unusually long evolutionary history' – please provide more context what are you comparing with? It would be nice to add context to what is currently known.

We have rephrased this sentence as follows. We specifically cite RLP42 and Cf-2 as examples where a broader comparative approach has been taken, but where the receptors seem to be present only in a single genus.

“Analysis of INR provides an example of a PRR with conserved recognition function across relatively long evolutionary history (since the emergence of the Phaseoloids ~28 mya), compared to other extensive studied LRR-RLPs which are generally studied within a single genus or species (40,41).”

Personally, I think the discussion could be made more concise.

We have simplified where possible (413-431, 433-440, 472-477).

326 – I would not agree that it represents an 'alternative' mechanism – I don't think there is currently a paradigm for how these receptors evolve. I think this work represents one of the few attempts to characterise the origin of the receptor in detail.

We thank the reviewer for this comment, and agree that additional case studies for receptors are needed to reveal broader patterns in PRR evolution. We do not aim to draw a sharp contrast with Cf-2 but still would like to highlight its evolution within the Solanum genus. We changed the cited text fragment accordingly:

“This evolutionary pattern is reminiscent of the evolution of LRR-RLP Cf-2 which evolved <6 mya within a genus Solanum, potentially by intergenic recombination (41). In summary, an ancestral gene insertion event is the likely source of extant gene and copy number variation at the INR locus, which preceded the evolution of a specific peptide recognition function.”

349 – gives the impression that EFR/XPS1/CORE is RLPs – rephrase. I would also argue that for some of the receptors there has been some extensive phenotyping than others – e.g. for RLP42.

We thank the reviewer for the helpful feedback on the paragraph regarding receptor evolution. We have re-written the paragraph to focus on examples where phenotyping has been performed and comparative genomic analysis could be helpful to define evolutionary patterns. We removed mention of RK for simplicity.

“Our detailed analysis of the emergence of INR provides a roadmap for understanding evolution of recognition specificity for other lineage-specific PRRs. For analogous RLP-mediated responses within the Brassicaceae, extensive phenotyping was earlier performed for several PAMPs -- eMax, nlp20, SCFE1, IF1 and pg13 – across Arabidopsis accessions and related species (27,28,33,63,64). However, analysis of the respective PRR loci across Brassicaceae has not yet been conducted, and may reveal processes leading to their dynamic evolution.”

356 – I would argue that this paper suggests that RLPs have dynamic evolutionary patterns – similar to NLRs, unlike RKs (in general). They highlight two examples that are conserved, but I would not say that these examples are representative.

The comments about complex vs conserved RLPs in the Arabidopsis pangenome are very interesting. We agree that Pruitt 2021 data (https://doi.org/10.1038/s41586-021-03829-0) did indeed show that many RLPs show rapid evolving patterns, just like NLRs. We have rewritten the paragraph to highlight the broader finding, while still pointing out these specific exceptions.

“A recent pangenomic analysis of Arabidopsis indicates that while many RLPs occur in complex or copy-number variable loci (18,27), three PAMP sensing LRR-RLPs (RLP 23, RLP30 and RLP32) are conserved across *A. thaliana* varieties (17). Further cross-species analysis may reveal a similar trajectory to INR, via ancestral duplications preceding fixation in the Arabidopsis lineage. In summary, additional case studies for specific receptors are needed to reveal broader patterns in PRR evolution.”

I would prefer that P-values are also shown on the graphs (potentially in addition to the notation of significance).

We opted to add the p-values and other statistical information in an extra supplementary for aesthetic reasons (supplementary table 7).

I am not qualified to comment on the quality of the sequencing – but I believe it represents a valuable resource of the community and is sufficient to support the claims in the manuscript.

We thank reviewer 2 for the appreciation of the assemblies provided in the manuscript.

364 – I would consider TMM a ligand binding LRR-RLP (https://www.ncbi.nlm.nih.gov/pmc/articles/PMC5458759/) although I appreciate it is significantly

We agree with the reviewer and changed the sentence accordingly:

“Despite multiple intense attempts, and in contrast to LRR-RKs (11,69), structural information for PAMP or HAMP sensing LRR-RLPs is not available, including INR (27,40,42).”

I would recommend citing this paper for the lack of RLP structure https://www.ncbi.nlm.nih.gov/pmc/articles/PMC6503760/.

We added the reference as suggested by reviewer 2 in both the introduction and the discussion.